# Breast cancer cell-secreted miR-199b-5p hijacks neurometabolic coupling to promote brain metastasis

Xianhui Ruan[1], Wei Yan[1], Minghui Cao[1], Ray Anthony M. Daza[1], Miranda Y. Fong[1,2], Kaifu Yang[3], Jun Wu[4], Xuxiang Liu[2], Melanie Palomares[5], Xiwei Wu[6], Arthur Li[7], Yuan Chen[8,9], Rahul Jandial[10], Nicholas C. Spitzer [11,12], Robert F. Hevner[1] & Shizhen Emily Wang [1,9] ✉

Breast cancer metastasis to the brain is a clinical challenge rising in prevalence. However, the underlying mechanisms, especially how cancer cells adapt a distant brain niche to facilitate colonization, remain poorly understood. A unique metabolic feature of the brain is the coupling between neurons and astrocytes through glutamate, glutamine, and lactate. Here we show that extracellular vesicles from breast cancer cells with a high potential to develop brain metastases carry high levels of miR-199b-5p, which shows higher levels in the blood of breast cancer patients with brain metastases comparing to those with metastatic cancer in other organs. miR-199b-5p targets solute carrier transporters (*SLC1A2*/EAAT2 in astrocytes and *SLC38A2*/SNAT2 and *SLC16A7*/MCT2 in neurons) to hijack the neuron–astrocyte metabolic coupling, leading to extracellular retention of these metabolites and promoting cancer cell growth. Our findings reveal a mechanism through which cancer cells of a non-brain origin reprogram neural metabolism to fuel brain metastases.

Metastasis is the leading cause of mortality in cancer patients. For breast cancer (BC), ~50% of patients treated with chemo- and/or hormone therapies develop distant metastases[1]. Metastatic BC (MBC) patients have a median survival of 1–2 years[2]. Symptomatic brain metastases are reported in 10–16% MBC, but the true incidence is higher, given the detection in 30% of patients at autopsy[3]. Brain metastases are more common for hormone receptor-negative, HER2[+] tumors or triple-negative (TN) tumors[4] and especially after chemotherapies or trastuzumab treatment, partially due to the blood-brain barrier (BBB) which cannot be crossed by most therapeutic

agents. Therefore, as systemic therapy of MBC improves, BC metastasis to the brain is expected to become more prevalent as cancer cells increasingly migrate to the "sanctuary" brain niche to escape therapy[5]. Little is known about the mechanisms of brain metastasis and current treatment, mainly radiotherapy and surgical resection, shows limited efficacy for brain-homing MBC.

Recent studies find that adaptation of a distant niche during pre-/metastatic stages, as a result of cancer cell-derived extracellular vesicles (EVs) and other secreted factors, is critical for the development of metastases[6]. Meanwhile, brain parenchymal cells have shown an

[1]Department of Pathology, University of California San Diego, La Jolla, CA, USA. [2]Department of Cancer Biology, City of Hope Beckman Research Institute, Duarte, CA, USA. [3]School of Biological Sciences, University of California San Diego, La Jolla, CA, USA. [4]Center for Comparative Medicine, City of Hope Beckman Research Institute, Duarte, CA, USA. [5]Cancer Prevention Movement, Arcadia, CA, USA. [6]Department of Computational and Quantitative Medicine, City of Hope Beckman Research Institute, Duarte, CA, USA. [7]Division of Biostatistics, City of Hope Beckman Research Institute, Duarte, CA, USA. [8]Department of Surgery, University of California San Diego, La Jolla, CA, USA. [9]Moores Cancer Center, University of California San Diego, La Jolla, CA, USA. [10]Department of Surgery; City of Hope, Duarte, CA, USA. [11]Neurobiology Department, School of Biological Sciences and Center for Neural Circuits and Behavior, University of California San Diego, La Jolla, CA, USA. [12]Kavli Institute for Brain and Mind, University of California San Diego, La Jolla, CA, USA. ✉e-mail: emilywang@ucsd.edu

emerging role in promoting primary and metastatic brain tumors by secreting pro-cancerous, pro-angiogenic, or immuno-modulating cytokines or EVs[7,8]. EV cargo, including miRNAs, can be transferred to distant cells to modulate cell functions[9–12]. Circulating miRNA has also emerged as potential biomarkers for cancer diagnosis and prognosis[13,14]. Here we focus on the miRNA cargo of BC-derived EVs for their role in adapting brain niche cells to facilitate brain metastasis.

In the brain, synaptic activity is tightly coupled with directional flows of metabolites between neurons and astrocytes[15]. Astrocytes enable rapid removal of glutamatergic neuron-secreted glutamate—an excitatory neurotransmitter—from the synaptic cleft by converting glutamate to glutamine, which is transported back to neurons for sustained neuronal activities[16]. This not only allows efficient replenishment of glutamine in glutamatergic neurons, but also protects neurons from the highly excitotoxic effect of prolonged glutamate exposure. Meanwhile, astrocytes also produce lactate from glycolysis and transport it to neurons[17]. Because glutamate inhibits glucose transport into the neurons, it is believed that lactate is a necessary energy substrate to fuel neurons and mediates a net energy transfer from astrocytes to neurons[15]. Under energy-deprived conditions, such as hypoglycemia and during intense activity of the nervous system, astrocytes use glycogen supplies to generate lactate as a way to protect neurons and ensure preservation of neuronal function[18]. The glutamate–glutamine cycle and lactate shuttle are further coordinated to couple neuronal activity to energy metabolism. Increased extracellular glutamate resulting from the activity of glutamatergic neurons stimulates glucose uptake and glycolysis in astrocytes, leading to increased production of lactate to be transferred to neurons[19,20]. Therefore, the neuron–astrocyte metabolic coupling featuring the unique intercellular transports of glutamate, glutamine, and lactate is critical to synaptic activity and normal brain functions, and is dynamically modulated during behavioral and pathological conditions such as learning and memory, sleep deprivation, neuroinflammation, and neurodegeneration[21].

In this work, we identify miR-199b-5p (hereinafter referred to as miR-199b) as a BC-derived regulator that impairs the metabolic coupling between neurons and astrocytes. We show that this effect of miR-199b leads to extracellular retention of glutamate, glutamine, and lactate, which in turn promotes the growth of BC cells metastasizing to the brain.

## Results

### miR-199b's association with BC brain metastasis

We started by profiling serum miRNAs from 42 patients with stage IV BC by small RNA-seq. These patients include 21 with brain metastases (cases) and 21 controls without brain metastases (but with metastases to other organs such as bone) that were selected to have matched age, time from diagnosis, and HER2/ER/PR status of the breast tumor. By comparing between the two groups, we identified four circulating miRNAs (miR-199b, miR-31-5p, miR-1299, and miR-206) that were relatively abundant and significantly higher in patients with brain metastases (fold change ≥ 2; $P < 0.05$) with average counts per million (CPM) ≥ 10 (Fig. 1a and Supplementary Data 1).

We next compared the EV and cellular miRNA profiles between parental MDA-MB-231 MBC cell line and a brain-tropic subline (MDA-231-BR3) selected in vivo[22]. EVs from the brain-tropic subline contained a higher level of miR-199b and a lower level of miR-31-5p comparing to the parental cells, whereas the other two pre-selected circulating miRNAs were not detected (Fig. 1b and Supplementary Data 2). miR-199b, miR-31-5p, and miR-1299 were detected from cellular RNA, but showed no significant difference between the parental and brain-tropic MDA-MB-231 (Fig. 1b and Supplementary Data 2). Based on the combined results of circulating miRNA profiling in MBC patients and EV miRNA profiling using brain-tropic MBC cells, we chose to focus on miR-199b for subsequent studies. A previous study reported that miR-

199 is uniquely upregulated in metastatic brain tumors compared to primary brain tumors[23], further supporting a role of this miRNA in brain metastasis.

Levels of circulating miR-199b were only associated with metastases to the brain, but not bones, in the same cohort of MBC patients (Fig. 1c). In contrast to the high miR-199b secretion by brain-tropic MDA-MB-231, a bone-tropic subline and a lung-tropic subline of MDA-MB-231 did not exhibit higher EV miR-199b levels comparing to parental MDA-MB-231 cells and MCF10A non-cancer cells (Fig. 1d). In addition, a brain-tropic subline of T47D BC cells as well as two independent BBM lines derived from brain metastases of BC patients that form brain-tropic metastases in mice[24,25], one HER2+ and one TN, all showed very high secretion levels of miR-199b (Fig. 1d). The higher miR-199b secretion levels in brain-tropic BC cells were not always accompanied by higher intracellular levels (Fig. 1d). These results collectively suggest that BC cell-derived, EV-encapsulated miR-199b partakes in the reprogramming of target cells to influence brain metastasis.

Nanoparticle tracking analysis (NTA) of MDA-MB-231- and BBM-derived EVs showed a typical size distribution of small EVs ranging from ~60–200 nm in diameter (Supplementary Fig. 1a). Several EV marker proteins, but not a Golgi marker, were detected in these EVs (Supplementary Fig. 1b). Iodixanol density gradient ultracentrifugation indicated co-enrichment of EV marker proteins and miR-199b (Supplementary Fig. 1b). For EVs from the brain-tropic MDA-231-BR3 and BBM cells, levels of miR-199b were not affected by treating EVs with protease followed by RNase to remove RNA outside of the EVs[26], but were significantly reduced when the membrane protection was destroyed (Supplementary Fig. 1c), indicating miR-199b is encapsulated in the lumen of EVs. For EVs from MDA-MB-231 cells engineered to overexpress and secrete high levels of exogenous miR-199b, partial protection by membranes was seen (Supplementary Fig. 1c), suggesting these cells employ more than one mode of miR-199b secretion leading to both extra-EV and intra-EV localization.

### Effect of EV miR-199b on metabolite influxes

To determine the potential function of miR-199b, we interrogated the microRNA.org and RNA22 algorithms, which predicted several plasma membrane transporters in the solute carrier (SLC) family as targets of miR-199b in both human and mouse. These include genes code for EAAT2 (*SLC1A2/GLT-1*) that controls >90% of glutamate influx into astrocytes[27], as well as SNAT2 (*SLC38A2/ATA2/SAT2*) and MCT2 (*SLC16A7*), the primary transporters for influxes of glutamine and lactate, respectively, into neurons[28,29]. *EAAT2*, *SNAT2*, and *MCT2* genes each carries two miR-199b binding sites in the 3′UTR. We cloned each 3′UTR into a luciferase reporter, using the wild-type 3′UTR or with one or both miR-199b sites mutated, to determine the responsiveness to miR-199b. For all three genes, both miR-199b sites in the 3′UTR turned out to participate in miR-199b's suppressive effect on target gene expression (Fig. 2a). Lastly, transfection of a miR-199b mimic, but not a control mimic, significantly suppressed the endogenous expression of EAAT2 in normal human astrocytes (NHA) as well as SNAT2 and MCT2 expression in differentiated SH-SY5Y neuron model at both RNA and protein levels (Fig. 2b, c).

Using human NHA and differentiated SH-SY5Y neuron models, we confirmed that these brain cells took up fluorescently labeled EVs under the culture conditions (Fig. 3a). Compared to the control treatment with MCF10A EVs or PBS, only EVs from brain-tropic MDA-231 or MDA-231 overexpressing and secreting miR-199b but not an irrelevant miRNA miR-211 (Supplementary Fig. 2a, b) led to significant downregulation of EAAT2, SNAT2, and MCT2 in brain cells (Fig. 3b, c). In contrast, EVs from brain-tropic MDA-231 cells expressing anti-miR-199b miRNA inhibitor, compared to EVs from control MDA-231-BR3 cells, elevated the expression of these genes (Fig. 3b, c). Treatment with EVs from BBM cells expressing anti-miR-199b also led to higher

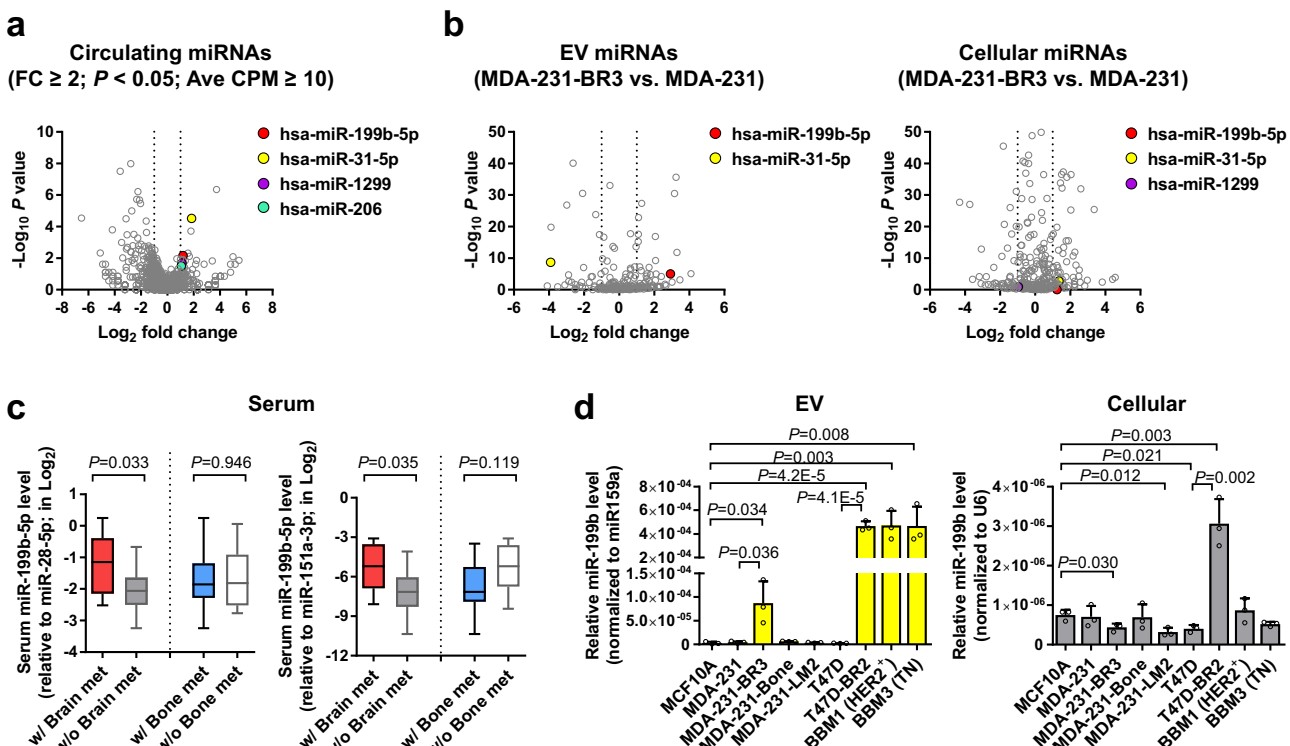

**Fig. 1 | miR-199b is associated with breast cancer brain metastasis. a, b** Volcano plots showing differentially expressed miRNAs in the sera of stage IV BC patients (**a**; case group with brain metastases vs. control group without; *n* = 21 for each group) and in the EV and cellular RNA of MBC cells (**b**; MDA-231-BR3 cells vs. parental cells). Data was from small RNA-seq. The positions of selected miRNAs were noted in each plot. **c** Relative levels of miR-199b in sera were determined by RT-qPCR, normalized to miR-28-5p or miR-151a-3p (both showing low coefficient of variation among all serum samples), and compared between stage IV BC patients with or without brain metastases, or between patients with or without bone metastases. For data using miR-28-5p as an internal reference, *n* = 9 w/ Brain met and *n* = 12 w/o Brain met, whereas *n* = 13 w/ Bone met and *n* = 8 w/o Bone met. For data using miR-151a-3p as

an internal reference, *n* = 9 w/ Brain met and *n* = 12 w/o Brain met, whereas *n* = 14 w/ Bone met and *n* = 7 w/o Bone met. **d** Relative levels of EV and cellular miR-199b were determined by RT-qPCR, normalized to ath-miR159a spike-in (for EV RNA) or U6 (for cellular RNA), and compared among indicated cell lines (*n* = 3 biological replicates). The boxes in the box-and-whiskers plots show the median (center line) and the quartile range (25–75%), and the whiskers extend from the quartile to the minimum and maximum values. In bar graphs, values are shown as mean ± SD. The edgeR package was used for the statistical test in **a** and **b**. Unpaired two-tailed *t*-test was used in **c** and **d**. *P* values are indicated. Source data are provided as a Source Data file.

expression of EAAT2, SNAT2, and MCT2 in brain cells compared to EVs from control BBM cells (Fig. 3c). Other glutamate transports, including EAAT1 (*SLC1A3/GLAST*) and ASCT2 (*SLC1A5*)[30], were not significantly regulated by EV miR-199b (Supplementary Fig. 2c). To obtain a comprehensive understanding of the expression patterns of EAAT2, SNAT2, and MCT2 in different types of brain cells, we re-analyzed a single cell gene expression dataset of human prefrontal cortex samples. Following cell clustering and cell type annotation, five major cell subsets were obtained (Supplementary Fig. 3a). Expression of EAAT2 was the highest in astrocytes, whereas expression of SNAT2 and MCT2 were the highest in excitatory neurons (Supplementary Fig. 3b, c). Oligodendrocytes also show some EAAT2 and MCT2 expression; however, in this study, we focus on astrocytes and neurons for their metabolic coupling pathway. We next measured glutamate consumption from the medium by astrocytes and glutamine and lactate consumption by SH-SY5Y neurons. When the brain cells were pre-treated with high-miR-199b EVs, consumptions of all three metabolites were suppressed (Fig. 4a). These effects were abolished when cells were pre-treated with EVs from MDA-231-BR3 expressing anti-miR-199b (Fig. 4a). To confirm the role of EV miR-199b-mediated regulation of EAAT2, SNAT2, and MCT2 in these effects, we overexpressed EAAT2 (in NHA), SNAT2 (in SH-SY5Y), and MCT2 (in SH-SY5Y) using plasmids carrying the cDNA of each target gene, and found that restoration of these targets blocked the effects of miR-199b on metabolite influxes (Fig. 4b–d).

We next adopted an ex vivo organotypic brain slice culture system[31–33] (Fig. 5a) to confirm that EVs can be taken up by neurons and astrocytes in mouse brain slice culture (Fig. 5b), and that EVs with high miR-199b levels (from miR-199b-overexpressing and brain-tropic MDA-MB-231 cells) downregulate *Eaat2*, *Snat2*, and *Mct2* expression in brain slices (Fig. 5c). Meanwhile, levels of extracellular glutamine and lactate remaining in the culture medium were significantly higher two days after high-miR-199b EV treatment (Fig. 5d), suggesting a suppressed net influx into the brain cells, although glutamate levels in the medium were below the detection threshold at this time point and culture condition. To further test if miR-199b-induced extracellular retention of glutamine and lactate affects cancer cell growth, we seeded fluorescently labeled MDA-231 or MDA-231-BR3 cells onto EV-treated brain slices and measured the fluorescence intensity three days later to evaluate cancer cell growth. EVs with varying levels of miR-199b did not directly affect the growth of these cancer cells in vitro (Supplementary Fig. 4a). For both cancer cell lines, enhanced growth of cancer cells was seen on brain slices treated with high-miR-199b EVs (Fig. 5e). This effect was diminished when we tested cancer cells with impaired ability to metabolize glutamine or lactate (by siRNA against *GLS* or *MCT1*; Fig. 5f and Supplementary Fig. 4b), indicating the growth-promoting effect was mediated by glutamine and lactate. In addition, we treated MDA-231 cancer cells with the conditioned medium (CM) collected from EV-treated brain slices, with or without supplementation with glutamine or/and lactate, and evaluated in vitro cancer cell

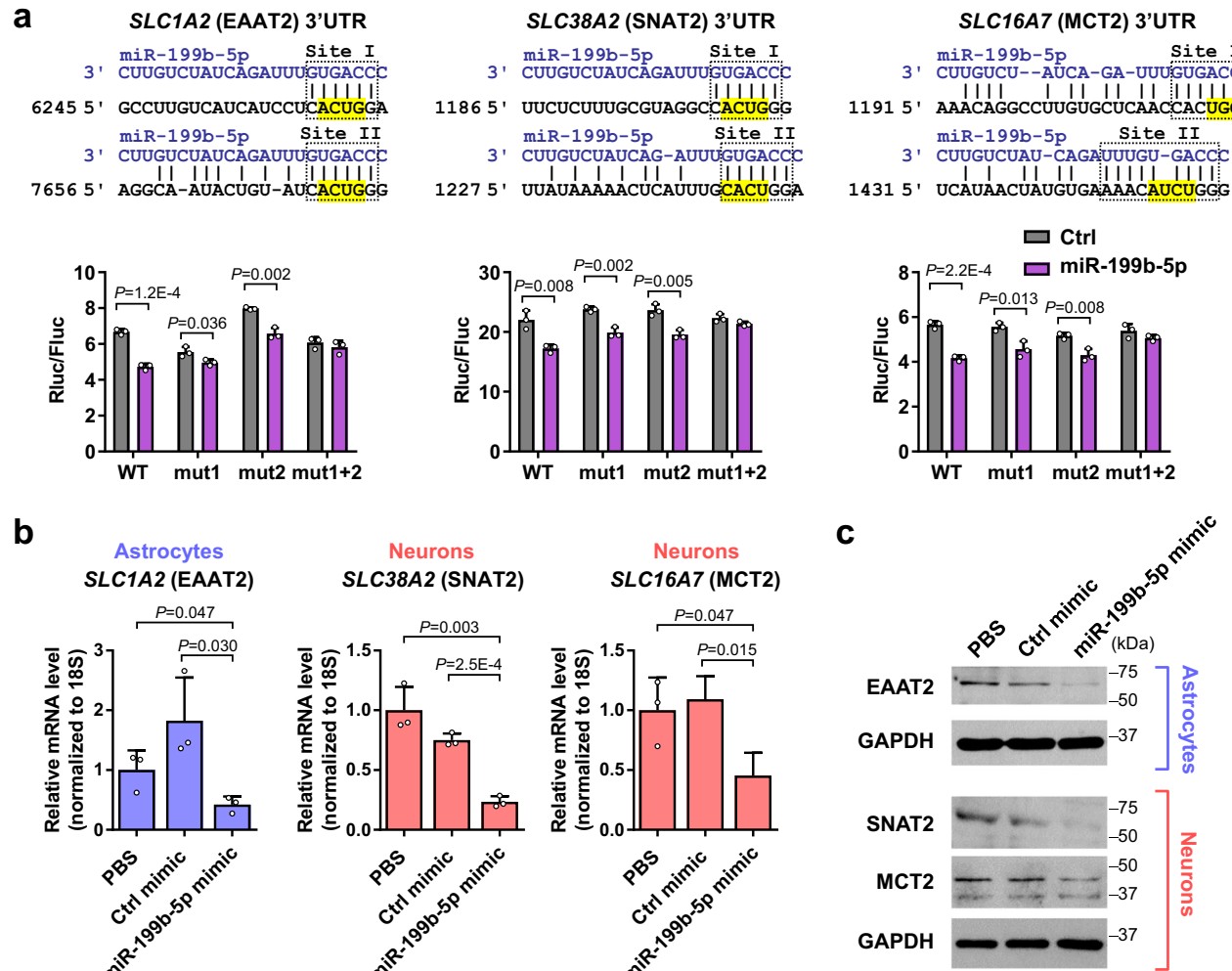

**Fig. 2 | miR-199b targets SLC transporters EAAT2, SNAT2, and MCT2.**
**a** Predicted miR-199b binding sites in the 3'UTR of human *SLC1A2* (EAAT2), *SLC38A2* (SNAT2), and *SLC16A7* (MCT2). The corresponding wild-type (WT) sequences are shown. For each gene, a 3'UTR fragment encompassing both miR-199b binding sites was cloned into a reporter plasmid. Highlighted sequences were mutated to destroy site I (mut1), site II (mut2), or both (mut1 + 2) in the reporter construct. Responsiveness of the reporters to miR-199b was determined in MDA-MB-231 cells overexpressing miR-199b or control ($n = 3$ biological replicates). **b** RT-qPCR-determined RNA levels of EAAT2 in astrocytes (NHA) and levels of SNAT2 and MCT2 in neurons (differentiated SH-SY5Y) upon transfection with miR-199b or control mimic. Data were normalized to 18S rRNA ($n = 3$ biological replicates). **c** Western blot showing levels of indicated proteins in astrocytes and neurons upon miRNA mimic transfection. GAPDH was used as a loading control. In bar graphs, values are shown as mean ± SD. Unpaired two-tailed *t*-test was used in **a** and **b**. *P* values are indicated. Source data are provided as a Source Data file.

growth in the CM. CM from brain slices treated with high-miR-199b EVs, which contained higher levels of glutamine and lactate, resulted in enhanced cancer cell growth compared to CM from brain slices treated with control EVs. Supplementation of the CM from control EV-treated brain slices with glutamine or lactate to match the levels detected in the CM from high-miR-199-treated groups partially rescue cancer cell growth, whereas adding both glutamine and lactate fully recapitulated the effects of high-miR-199b EVs (Fig. 5g). In the brain slice studies, EV treatment did not significantly affect brain cell viability during the experiment period (Supplementary Fig. 4c).

**Effect of miR-199b on BC brain metastasis**
To evaluate the in vivo function of BC-derived EVs in the brain, we first examined the uptake of intravenously (i.v.) injected MDA-MB-231 EVs by brain cells. We used immunofluorescence to detect human-specific CD63 indicative of human cell-derived EVs in both GFAP[+] astrocytes and MAP2[+] neurons (Fig. 6a), confirming the ability of circulating EVs to cross the BBB and influence brain cells. EVs carrying varying levels of miR-199b did not show a difference in the brain cell uptake efficiency

(Supplementary Fig. 5). Due to the expected degradation of EV-transferred human CD63 proteins in mouse brain cells, the actual EV uptake efficiency in the brain cells is expected to be higher than detected herein. EVs from MDA-MB-231 expressing miR-199b or control were i.v. injected into the tail vein twice a week for 5 weeks before the brain was collected. Treatment with high-miR-199b EVs resulted in higher levels of miR-199b (Fig. 6b) and lower expression levels of *Eaat2*, *Snat2*, and *Mct2* in brain tissues (Fig. 6c, d). We further separated the interstitial fraction from the cell-containing fraction following brain tissue dissociation and measured the levels of metabolites in the cell-free interstitial fraction. Brains from mice treated with high-miR-199b EVs showed higher levels of extracellular glutamine, glutamate, and lactate compared to control groups receiving saline or low-miR-199b EVs (Fig. 6e). In contrast, lung tissues from these mice did not show significant differences in the extracellular levels of glutamine and lactate (Fig. 6f), whereas interstitial glutamate could not be detected in the lungs. Next, in mice that have received EV treatment, we injected luciferase-labeled MBC cells into the mammary fat pad to monitor development of metastases in the brain. After 5 weeks, mice that had

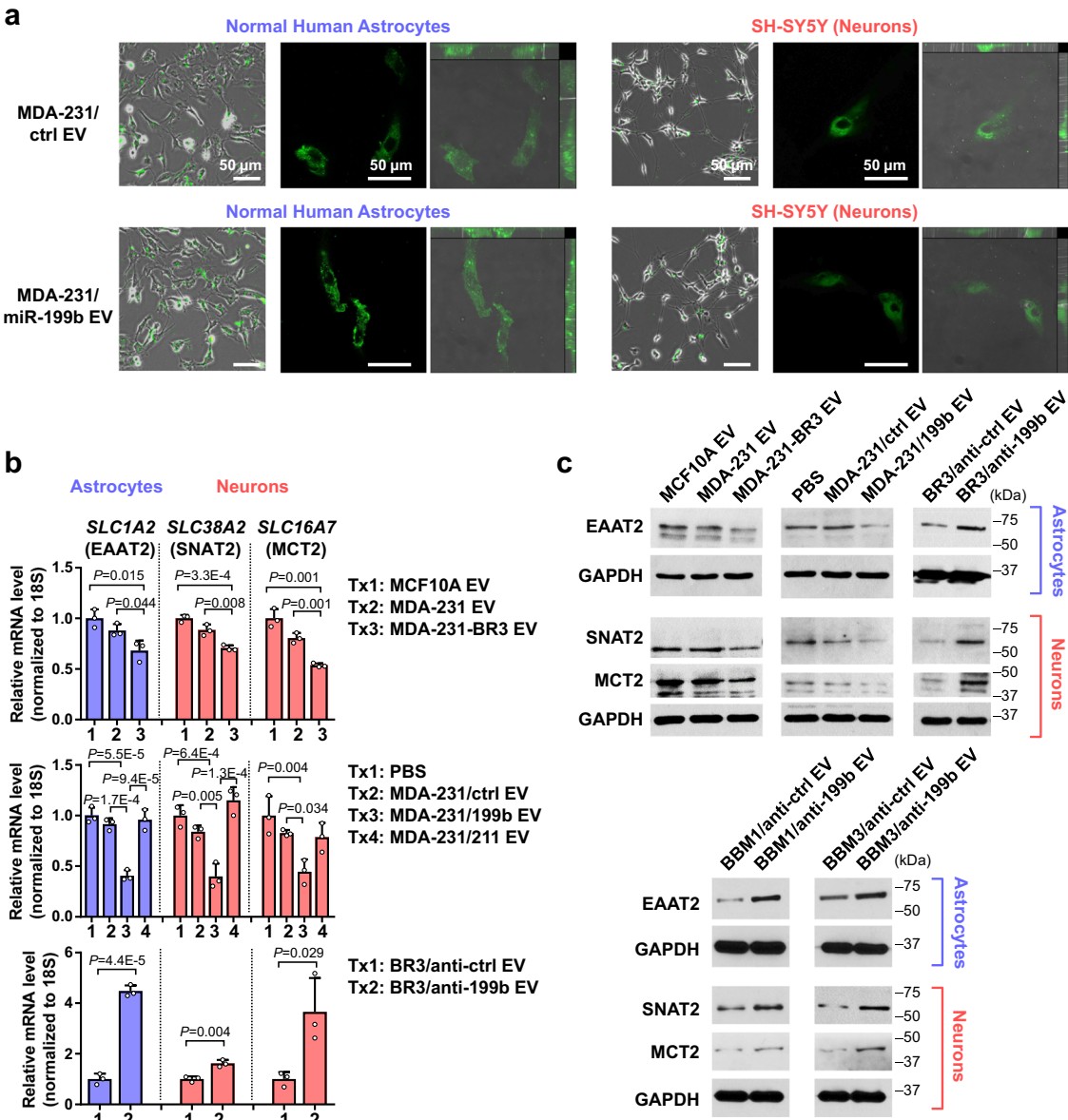

**Fig. 3 | EV miR-199b suppresses the expression of EAAT2, SNAT2, and MCT2 in astrocytes and neurons. a** Fluorescence (green) was detected in astrocytes (NHA) and neurons (differentiated SH-SY5Y) after an incubation with CFSE-labeled EVs for 24 h, which indicates EV uptake. A widefield image (CFSE and phase contrast), a confocal image (only showing CFSE), and orthogonal views of confocal microscopy z-stacks (CFSE and DIC) showing the 3D localization of CFSE signals are presented. **b** RT-qPCR-determined RNA levels of EAAT2 in astrocytes and levels of SNAT2 and MCT2 in neurons upon treatment with indicated EVs for 48 h. Data were normalized to 18S rRNA ($n = 3$ biological replicates). Values are shown as mean ± SD. **c** Western blot showing levels of indicated proteins in astrocytes and neurons upon treatment with indicated EVs for 48 h. GAPDH was used as a sample processing control. The samples derive from the same experiment but in some cases different gels for EAAT2, SNAT2, or MCT2 and another for GAPDH were processed in parallel. Unpaired two-tailed t-test was used in the top and bottom panels of **b**. One-way ANOVA followed by Tukey's multiple comparison test was used in the middle panel of **b**. P values are indicated. Source data are provided as a Source Data file.

received high-miR-199b EVs showed a higher degree of metastases developed in the brain (Fig. 6g). In contrast, growth of the primary mammary tumors did not show significant differences among all groups (Fig. 6h), suggesting the enhanced brain metastasis in high-miR-199b EV-treated group was not associated with a different primary tumor burden.

We next evaluated the effect of miR-199b on BC tumor growth in the brain. Luciferase-labeled MDA-MB-231 cells stably overexpressing miR-199b or control were intracranially injected into female mice to establish intracerebral tumor. The miR-199b-overexpressing tumor group showed enhanced tumor growth in the brain compared to the control group as detected by bioluminescence imaging (Fig. 7a, b). This is unlikely to be associated with a direct effect of miR-199b on

cancer cell proliferation, as the miR-199b-overexpressing cells showed a lower proliferation rate than control cells when cultured in vitro (Fig. 7c). To characterize potential changes in brain metabolism of glutamine, we infused mice bearing intracerebral BC tumors with $^{13}$C-labeled glutamine. Following whole brain dissociation, the cell-free interstitial fraction was analyzed by 2D NMR to quantify $^{13}$C-labeled metabolites. Higher extracellular levels of $^{13}$C-labeled glutamate, acetate, and valine and lower levels of GABA were detected in the brains bearing high-miR-199b tumors (Fig. 7d). This result is consistent with the hypothesized function of miR-199b to induce extracellular retention of glutamate, and indicates decreased GABA synthesis from glutamine, likely as a result of decreased glutamine uptake into neurons. When mCherry-labeled, brain-tropic MDA-MB-231 cells

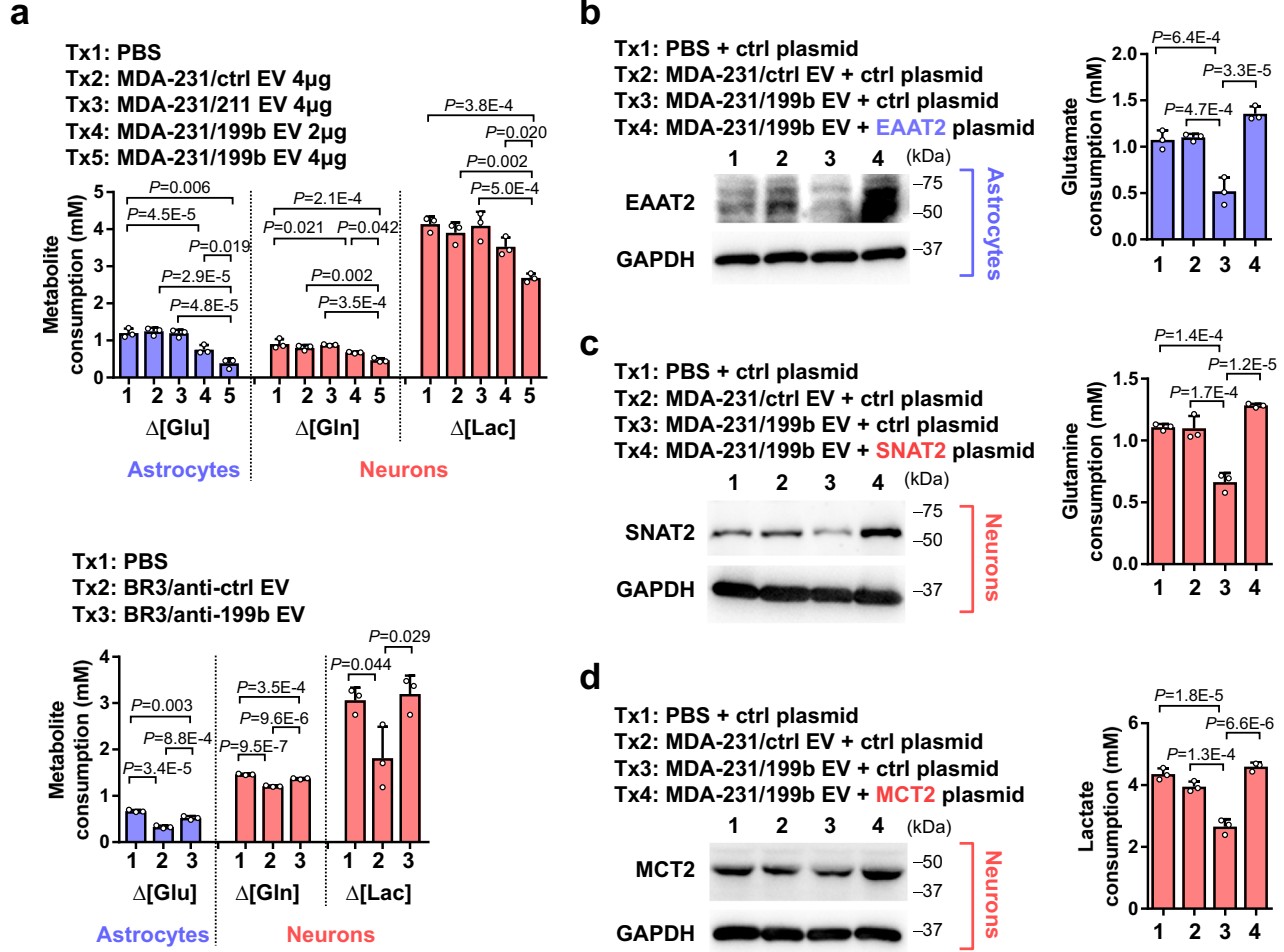

**Fig. 4 | EV miR-199b suppresses metabolite influxes in astrocytes and neurons.**
**a** Astrocytes and neurons were treated with indicated EVs for 48 h and then changed to the media described in Methods. Net consumption of each metabolite was determined by calculating the difference of metabolite levels in the medium at the start and end points. To determine glutamate consumption, a starting level of 3 mM glutamate was used and the remaining glutamate in the culture medium of astrocytes was measured 24 h later ($n = 3$ biological replicates). To determine glutamine consumption, a starting level of 2 mM glutamine was used and the remaining glutamine in the culture medium of neurons was measured 48 h later

($n = 3$ biological replicates). To determine lactate consumption, a starting level of 25 mM lactate was used and the remaining lactate in the culture medium of neurons was measured 12 h later ($n = 3$ biological replicates). **b–d** Astrocytes (**b**) or neurons (**c**, **d**) were transfected with an overexpressing plasmid encoding EAAT2, SNAT2, or MCT2, or with the vector as a control, before treatment with EVs. Metabolite assays were carried out as in a ($n = 3$ biological replicates). In bar graphs, values are shown as mean ± SD. One-way ANOVA followed by Tukey's multiple comparison test was used in **a**–**d**. $P$ values are indicated. Source data are provided as a Source Data file.

expressing anti-miR-199b or control were intracranially injected, tumors expressing anti-miR-199b showed significantly suppressed growth in the brain (Fig. 7e). Tracing of infused [13]C-labeled glutamine showed lower extracellular levels of [13]C-labeled glutamine, glutamate, and succinate and higher levels of GABA and threonine in the brains bearing anti-miR-199b tumors (Fig. 7f). In another experiment, we treated mice with i.v. injections of EVs from T47D BC cells with or without miR-199b overexpression and secretion (Supplementary Fig. 2a) prior to an intracardiac injection with T47D-BR2 cells. High-miR-199b EV treatment enhanced development of brain metastases in this model (Fig. 7g–i), but did not significantly affect metastases to lungs and liver (Supplementary Fig. 6). We also observed metastases to the ovaries (Supplementary Fig. 6), which could be related to ER signaling in this ER+ tumor model that requires administrations of estrogen. As such, results from the intracerebral tumor growth model, breast-to-brain metastasis model, and hematogenous metastasis model collectively support a role of BC cell-derived EV miR-199b in promoting brain metastasis.

## Discussion

Accumulating evidence indicates that cells at a distant site can take up tumor-derived EVs from the circulation, leading to altered cellular behaviors which, in turn, facilitate metastasis[34–36] or induce other systemic effects such as hyperglycemia and cachexia[37–39]. Here we focus on miR-199b for the specific association of circulating miR-199b with the presence of brain metastases (but not bone metastases) in BC patients, and for the high EV secretion of miR-199b by brain-tropic BC cells, including the in vivo selected MDA-MB-231 model and patient-derived BBM models. Interestingly, previous studies have identified several tumor-suppressive functions of the miR-199 family inside of cancer cells. miR-199b-5p targets HER2 and its downstream signaling in HER2-overexpressing BC cell lines[40]. The miR-199 family also suppresses migration and proliferation in head and neck and colon cancer cells but promote stem cell properties, metastasis, and drug resistance in other cancers[41–45]. Another study finds that miR-199b-5p is repressed in BC cells and is downregulated in vascular endothelial cells during angiogenesis[46]. How cancer cell-secreted, EV-encapsulated miR-199b

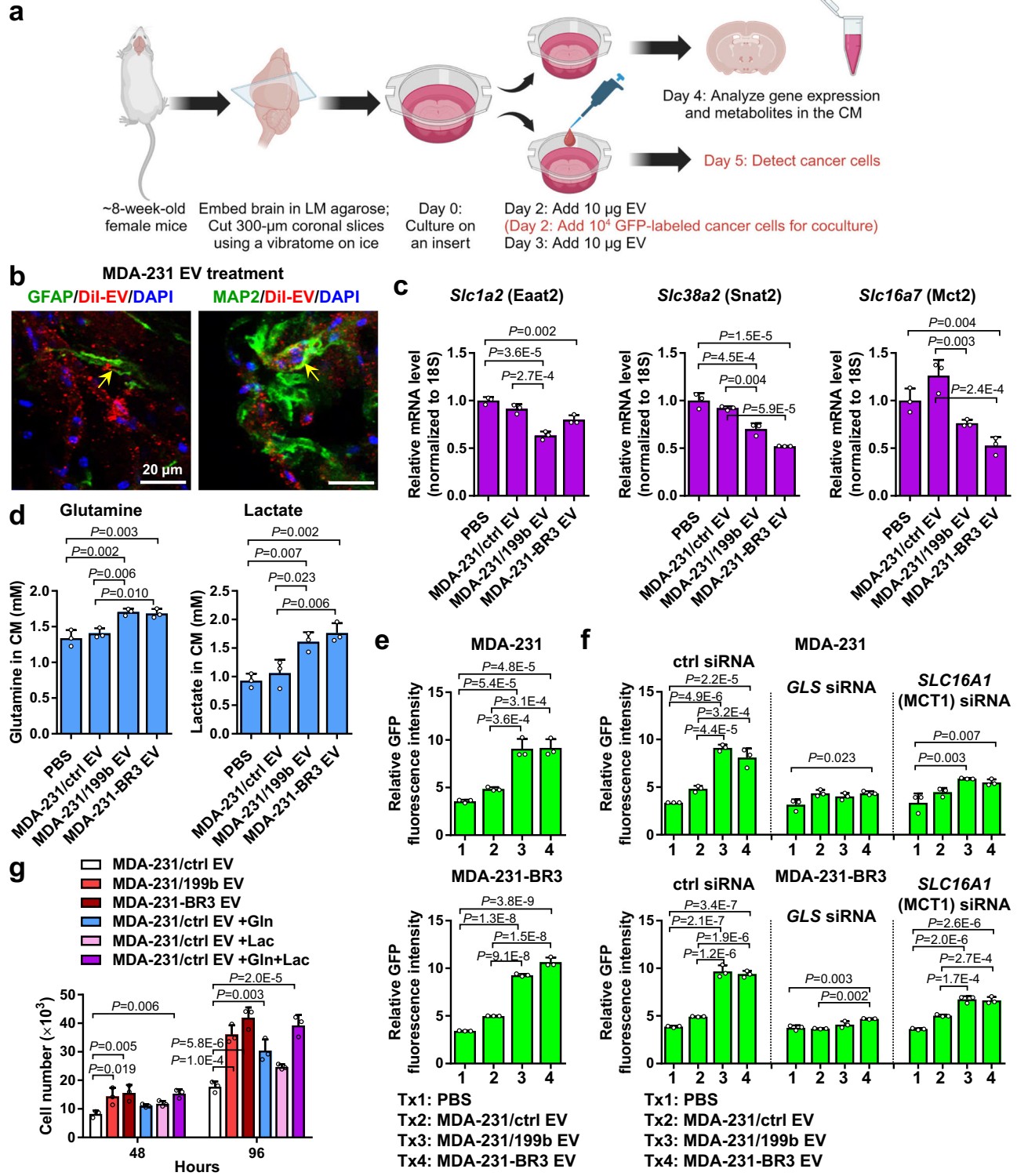

**Fig. 5 | EV miR-199b promotes BC cell growth on brain slices. a** Brain slice preparation and experiment procedure. Created with BioRender.com. **b** Brain slice uptake of EVs detected by IF of DiI-labeled EVs (red) in GFAP+ astrocytes and MAP2+ neurons (green). DAPI (blue) shows the nuclei. Experiment was repeated twice with similar results. **c** RT-qPCR-determined RNA levels of indicated genes in EV-treated brain slice (*n* = 3 biological replicates). **d** Levels of metabolites in the culture media determined by ELISA (*n* = 3 biological replicates). **e** GFP-labeled MDA-231 or MDA-231-BR3 cells were seeded onto the brain slices on day 2; total GFP fluorescence intensity on each brain slice was measured on day 5 to indicate the abundance of labeled cancer cells (*n* = 3 biological replicates). **f** GFP-labeled MDA-231 or MDA-231-BR3 cells transfected with indicated siRNA were seeded on day 2 and the fluorescence intensity on each brain slice was measured on day 5 (*n* = 3 biological replicates). **g** MDA-231 cells seeded in 12-well plates were cultured in CM collected from indicated EV-treated brain slice cultures. When indicated, glutamine and lactate were added to the CM to reach a final concentration of 1.71 mM and 1.61 mM, respectively. Cell numbers were counted after 48 and 96 h (*n* = 3 biological replicates). In bar graphs, values are shown as mean ± SD. One-way ANOVA followed by Tukey's multiple comparison test was used in **c**–**g**. *P* values are indicated. Source data are provided as a Source Data file.

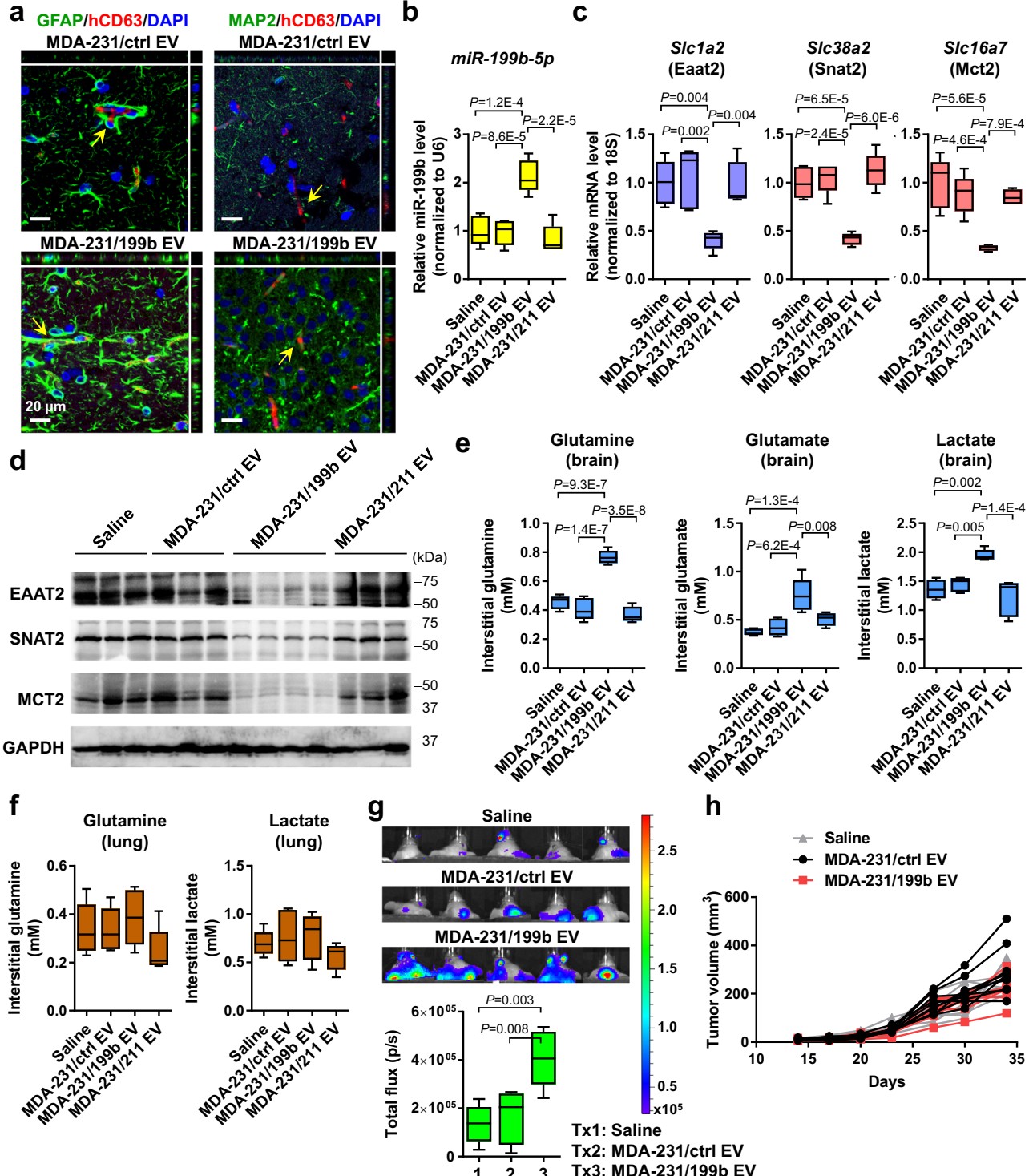

influences normal cells remained unclear. Of note, we find that the brain-tropic BC cells used in this study do not contain a higher intracellular level of miR-199b compared to normal cells and non-brain-tropic cancer cells, suggesting the miRNA could exert a function outside of the secreting cancer cell.

Adaptations of various cells in the brain to establish a premetastatic niche that facilitates BC brain metastasis may require the combined actions of multiple EV cargoes. Tominaga et al. find that miR-181c in EVs secreted by brain metastatic cancer cells targets 3-phosphoinositide-dependent protein kinase-1 to dysregulate actin dynamics and promote the destruction of BBB[47]. Our group has

previously reported that BC cells secrete miR-122 to suppress glucose metabolism in normal cells, which favors the use of glucose by cancer cells and promotes distant metastases[48]. Meanwhile, others have reported that primary and metastatic brain tumors can become less dependent on glucose by co-oxidizing acetate[49] or, in the case of BC brain metastases, by enhancing gluconeogenesis (which can initiate from lactate) and glutamine oxidation[50]. Our current study addressed how use of brain-enriched metabolites, especially glutamine, glutamate, and lactate, is reprogrammed in brain cells as an effect of BC-derived EVs to facilitate growth of brain metastases (Supplementary Fig. 7). In the brain tissues of mice bearing high-miR-199b-secreting

**Fig. 6 | EV miR-199b regulates metabolite transporters in the brain and promotes breast-to-brain metastasis. a** EVs from indicated cells were i.v. injected into NSG mice. After 24 h, the brain was collected and sectioned for IF analysis to detect human-specific CD63 (red) in GFAP+ astrocytes and MAP2+ neurons (green). DAPI (blue) shows the nuclei. Orthogonal views of confocal microscopy z-stacks are shown. Experiment was repeated twice with similar results. **b, c** NSG mice received i.v. injections of indicated EVs twice a week for 5 weeks. After the last EV treatment, the brain was collected and RNA levels of miR-199b and indicated genes were determined by RT-qPCR. Data were normalized to U6 (for miR-199b) or 18 S rRNA (for mRNAs) (n = 5 mice per group). **d** Western blot showing levels of indicated proteins in brain tissues from EV-treated mice. GAPDH was used as a loading control. Experiment was repeated twice with similar results. **e, f** After EV treatment, brain (**e**) and lung (**f**) tissues were dissociated, and the cell-free (interstitial) fraction was separated and measured for extracellular levels of glutamate, glutamine, and lactate (n = 5 mice per group, except for the Saline group in **e** Lactate measurement with n = 4 mice). Glutamate levels in lung tissues were below the limit of detection. **g** After 5 weeks of EV treatment, mice received mammary fat pad injection of luciferase-labeled MDA-MB-231-HM cells. Bioluminescent imaging and quantification results at week 5 after cancer cell injection were shown (n = 5 mice per group). **h** Measurements of the primary mammary tumor volume at the indicated time points (n = 10 mice per group). The boxes in the box-and-whiskers plots show the median (center line) and the quartile range (25–75%), and the whiskers extend from the quartile to the minimum and maximum values. One-way ANOVA followed by Tukey's multiple comparison test was used in **b, c, e, f,** and **g**. Two-way ANOVA was used in h (repeated measures). P values are indicated. No statistical significance was found in **f** and **h**. Source data are provided as a Source Data file.

intracranial MDA-MB-231 tumors, we detected higher levels of [13]C-labeled, extracellular glutamate and acetate and lower levels of GABA following [13]C-glutamine infusion. This result could reflect a combined effect of decreased brain cell uptake of glutamine and hence decreased production of glutamine-derived GABA, as well as decreased brain cell uptake of glutamate and acetate. As a mono-carboxylate and similar to lactate, acetate is also transported via monocarboxylate transporters and its influx into brain cells could also be suppressed upon MCT2 downregulation, which could lead to a higher extracellular level. Whether the overall acetate level (in contrast to [13]C-glutamine-derived acetate detected here) and its influx alter in the brain and how that potentially affects cancer cell metabolism and tumor growth remain to be studied. The critical effect of miR-199b on the dysregulated transports of glutamine, glutamate, and lactate seems to be unique to the brain due to the neuron–astrocyte coupling, as it did not alter the interstitial levels of glutamine and lactate in the lungs. Previous studies have revealed complex tumor regulatory effects of glutamine and lactate that are independent of their function to fuel cancer cell metabolism. In some tumor models, dietary gluta-mine supplementation inhibits tumor progression through suppres-sing epigenetically-activated oncogenic pathways, whereas glutamine restriction promotes tumor growth[51,52]. Lactate in the tumor micro-environment has been shown to promote tumor growth and pro-gression through metabolic and functional regulations of various immune cell subsets[53,54]. Whether these non-metabolic functions of glutamine and lactate also influence brain metastasis remains to be elucidated.

Recent studies have established a key role of astrocytes in the reprogramming of primary and secondary brain tumor micro-environment. Priego et al. report that brain metastatic cells induce the signal transducer and activator of transcription 3 pathway in a sub-population of reactive astrocytes surrounding metastatic lesions, which further influences the innate and acquired immune system to promote the viability of brain metastasis[55]. Tumor-associated astro-cytes shape the metabolic and immune compartments in the brain tumor microenvironment through the production of cholesterol and pro-inflammatory cytokines[56,57]. In addition, some cancer cells pro-mote the assembly of carcinoma-astrocyte gap junctions, which are used as channels to transfer the second messenger 2'3'-cyclic GMP-AMP to astrocytes, inducing production of inflammatory cytokines in the latter and supporting tumor growth in the brain[58]. Targeting the pro-cancerous reprogramming of astrocytes has thus emerged as a promising therapeutic strategy to treat and prevent primary and metastatic brain tumors. Our findings herein further support astro-cytes as a target for adjuvant therapies to prevent brain metastasis and restore normal brain metabolism.

Development of pharmacological agents targeting brain metas-tases is largely hindered by the limited BBB permeability of many anti-tumor compounds and their potential neurotoxicity. For example, although targeting tumor metabolism of glutamine by GLS inhibition could theoretically block the herein studied mechanism of metabolic reprogramming, the clinically applicable GLS inhibitor CB-839 has very low BBB permeability[59]. Inhibitors of GLUD1 (to block glutamate metabolism) are highly toxic to the brain[60], whereas the lactate flux inhibitor AZD3965 in several anti-cancer clinical trials targets both MCT1 and MCT2 so will also impair brain metabolism[61]. Our current mechanistic study suggests opportunities to block cancer EV-induced metabolic reprogramming of brain cells, which may protect normal brain function and meanwhile suppress brain metastases. For example, BBB-penetrating compounds that can increase the expression of miR-199b-targeted SLC transporters could potentially block the brain-adapting effect of BC-derived EVs. Ceftriaxone, a β-lactam antibiotic, exerts strong neuroprotective effects by increasing EAAT2 expression to prevent glutamate neurotoxicity[62,63] and is in clinical trials for neu-rological disorders such as Parkinson's disease[64]. Forskolin, a diterpe-noid isolated from roots of the plant Coleus forskohlii and a cAMP inducer, potently increases SNAT2 expression[65,66] and has been pro-posed as a neuroprotective agent. If these neuroprotective com-pounds suppress primary and metastatic brain tumors warrants further investigation.

## Methods

This research complies with all relevant ethical regulations. All animal experiments were approved by the institutional animal care and use committee at the University of California San Diego and City of Hope Beckman Research Institute. All research with human specimens were approved by the institutional review board committees at UC San Diego Health and City of Hope National Medical Center. All cells and clinical specimens used in this study were female, as ~99% of breast cancer occurs in women.

### Cells and constructs

Cells were cultured at 37 °C in a humidified incubator with 5% $CO_2$. MCF-10A (CRL-10317; female), MDA-MB-231 (HTB-26; female), T47D (HTB-133; female), and SH-SY5Y (CRL-2266; female) cells were obtained from American Type Culture Collection (ATCC; Manassas, VA). MCF-10A cells were cultured as described[67]. MDA-MB-231 cells were cultured in Dulbecco's Modified Eagle's medium (DMEM; Gibco, Waltham, MA) supplemented with 10% fetal bovine serum (FBS; Sigma-Aldrich, St. Louis, MO). T47D cells were cultured in RPMI-1640 medium (Gibco) supplemented with 10% FBS. The brain-tropic subline MDA-MB-231-BR3 was developed from the MDA-231BR brain-seeking clone previously generated by Dr. Yoneda et al.[22] To further enhance brain tropism, MDA-231BR cells were inoculated into the left ventricle of the heart in female NSG mice and brain metastases were isolated and cultured in vitro to generate MDA-MB-231-BR2, which subsequently underwent another round of in vivo selection and explant culture to generated MDA-MB-231-BR3 from the brain metastases. The bone-tropic subline of MDA-MB-231 (MDA-231-Bone) was previously gener-ated by Dr. Yoneda et al.[22] and used in our previous study[68]. The lung-metastatic MDA-MB-231 (MDA-231-LM2)[69] was a gift from Drs. Yibin Kang and Hanqiu Zheng. A brain-tropic subline of T47D, namely

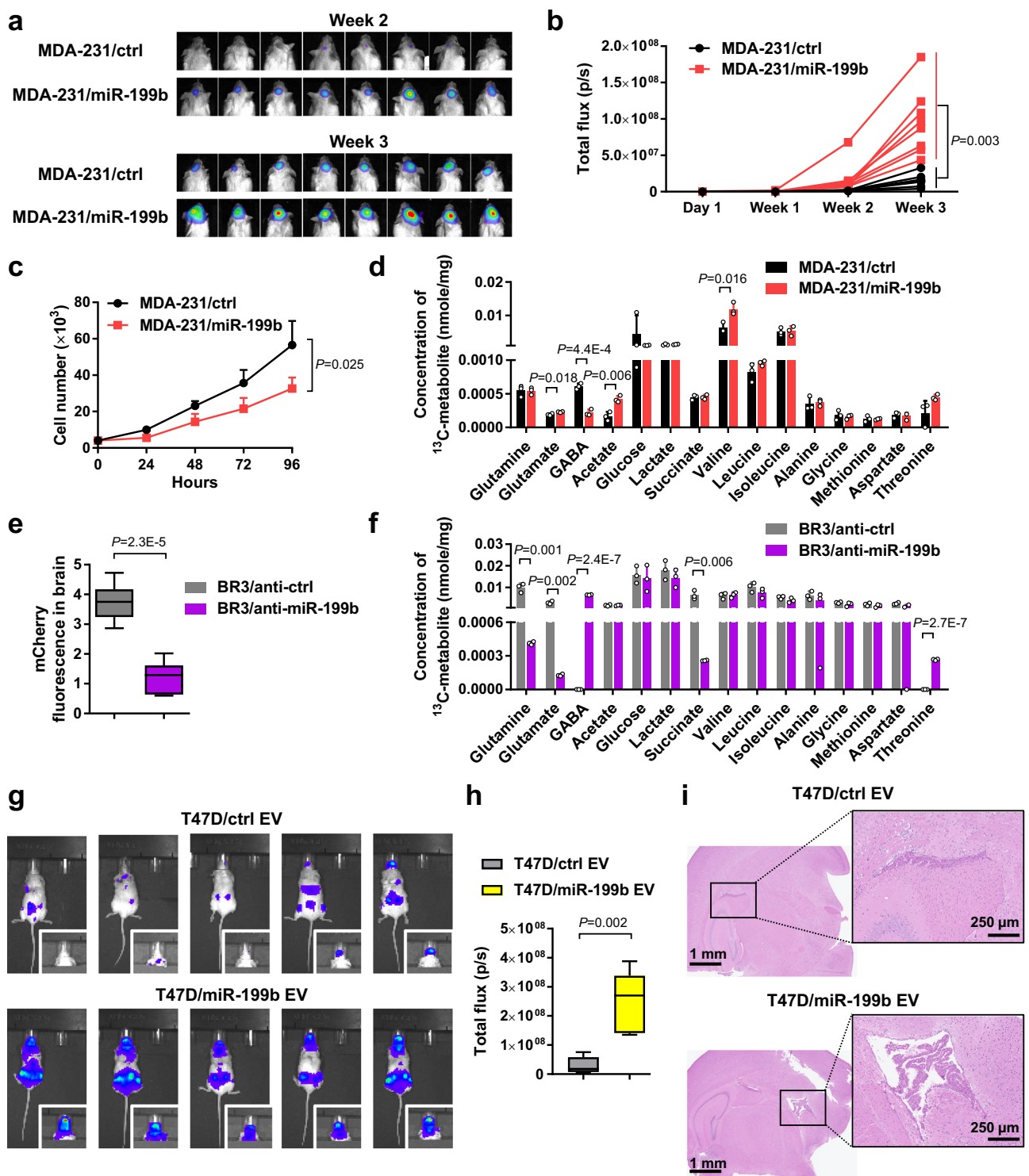

T47D-BR2, was generated from a spontaneous brain metastasis of T47D using the same in vivo selection strategy and was a gift from Dr. Yumei Feng. SH-SY5Y cells were maintained in DMEM/F-12 supplemented with 10% FBS and antibiotic-antimycotic (Gibco). Differentiation was induced by culturing SH-SY5Y cells in DMEM/F-12 supplemented with 2% FBS, antibiotic-antimycotic, and 10 μM all trans-retinoic acid (Sigma-Aldrich) for 7 days. A neuron-like phenotype was confirmed by expression of MAP2, a neuronal marker, by Western blotting. Normal human astrocytes (NHA; female; lot # 0000565612) were obtained from Lonza (Basel, Switzerland) and cultured in astrocyte basal medium with SingleQuots supplements (Lonza) following

manufacturer's instructions. Patient-derived breast-to-brain metastatic (BBM) cells (female) were developed by Dr. Jandial's group from resected specimens[24,25]. BBM1 cells were from a HER2[+] specimen, whereas BBM3 cells were from a triple-negative specimen. BBM cells were cultured in Advanced DMEM/F12 (Gibco) supplemented with 10% FBS, 2 mM glutamine (Gibco), and antibiotic-antimycotic on plates pre-coated with collagen I (Gibco). All cells used herein were tested to be free of mycoplasma contamination.

MDA-MB-231 and T47D cells were engineered to stably overexpress hsa-mir-199b (MDA-231/199b and T47D/199b), hsa-mir-211 (MDA-231/211), or the control vector (MDA-231/ctrl and T47D/ctrl)

**Fig. 7 | miR-199b enhances the growth of BC tumors in the brain and alters glutamine metabolism. a**, **b** Luciferase-labeled MDA-MB-231/miR-199b or control cells were intracranially injected into NSG mice (*n* = 8 mice per group). Bioluminescent images at week 2 and week 3 (**a**) were quantified (**b**). **c** In vitro cell proliferation assessments by counting cell numbers every 24 h (*n* = 3 biological replicates). Medium was replenished every other day. **d** After the bioluminescent imaging at week 3, mice in a received infusion of $^{13}C$-labeled glutamine. The whole brain was dissociated and the cell-free interstitial fraction was analyzed by 2D NMR to quantify $^{13}C$-labeled metabolites. Data were normalized to the dry weight of interstitial fraction (*n* = 3 mice per group). **e** mCherry-labeled MDA-231-BR3/anti-miR-199b or control cells were intracranially injected into NSG mice. At week 3, the whole brain was dissociated and the cell-containing fraction was used for measurement of mCherry fluorescence (*n* = 6 mice per group). **f** In mice received intracranial injections of MDA-231-BR3/anti-miR-199b or control cells, $^{13}C$-labeled glutamine was administered by infusion at week 3 and the whole brain was collected and dissociated. The cell-free interstitial fraction was analyzed by 2D NMR to quantify $^{13}C$-labeled metabolites. Data were normalized to the dry weight of interstitial fraction (*n* = 3 mice per group). **g**, **h** EVs from T47D/ctrl and T47D/miR-199b cells were i.v. injected into NSG mice twice a week for 5 weeks. After the last EV treatment, mice received an intracardiac injection of luciferase-labeled T47D-BR2 cells. In vivo whole body bioluminescent imaging (**g**) and quantification of signals in the head (**h**) at week 5 after cancer cell injection were shown (*n* = 5 mice per group). **i** Representative brain H&E histological images showing brain metastases in mice from **g**. The boxes in the box-and-whiskers plots show the median (center line) and the quartile range (25–75%), and the whiskers extend from the quartile to the minimum and maximum values. In bar graphs, values are shown as mean ± SD. Two-way ANOVA was used in **b** and **c** (repeated measures). Unpaired two-tailed *t*-test was used in **d**, **e**, **f**, and **h**. *P* values are indicated. Source data are provided as a Source Data file.

using lentiviral constructs HmiR0152-MR03, HmiR0250-MR03, or pEZX-MR03 (vector) from GeneCopoeia (Rockville, MD). The pEZX-MR03 vector encodes an eGFP tag and was used to stably label cells with GFP when indicated. MDA-MB-231-BR3 and BBM cells were engineered to stably overexpress a miR-199b-5p inhibitor (BR3/anti-199b and BBM/anti-199b) or the control vector (BR3/anti-ctrl and BBM/anti-ctrl) using lentiviral constructs HmiR-AN0291-AM03 and pEZX-AM03 (GeneCopoeia). The pEZX-AM03 vector also encodes an mCherry tag for cell labeling. Hsa-miR-199b-5p mimic (Cat# C-300561-05) and negative control mimic (Cat# CN-001000-01) were obtained from Dharmacon (Lafayette, CO). The siRNAs targeting human *GLS* and *SLC16A1* (MCT1) as well as the AllStars negative control siRNA were obtained from Qiagen (Hilden, Germany; GeneGlobe ID SI03155019, SI04179602, and SI03650318).

To clone the wild-type (WT) 3′ UTR of human *SLC1A2* (EAAT2), *SLC38A2* (SNAT2), and *SLC16A7* (MCT2), PCR-amplified fragments were digested with XhoI and NotI and then inserted into the same sites of the psiCHECK-2 reporter vector (Promega, Madison, WI) downstream of the *Renilla* luciferase (Rluc) gene. The PCR primers are: 5′-GCACGTCCTCGAGCAGACCTCACTGCTTTGTAGAG-3′ and 5′-TTATAATGCGGCCGCTACTCCTACCACCATCTCA-3′ for *SLC1A2* (EAAT2), 5′-GCACGTCCTCGAGTATTCAGTTGCATCTGTCTCCA-3′ and 5′-CGATAATGCGGCCGCAAATGGACCAACGGTTTCAC-3′ for *SLC38A2* (SNAT2), and 5′-GCGCGTCCTCGAGAAAGTCTGCATGACTCTTAACAATG-3′ and 5′-TTCTGCTGCGGCCGCTCCTAAATATGAGCCACATGCT-3′ for *SLC16A7* (MCT2). Point mutations were introduced into the WT 3′ UTR constructs by using the following PCR primers (mutated nucleotides underlined): 5′-GCCTTGTCATCATCCTCGACAGAGATGGAG-3′ and 5′-GCAACATCTCCATCTCTGTCGAGGATGATGACAAGGC-3′ for *SLC1A2* (EAAT2) mut1, 5′- GAACCAACTATGAGGCAATACTGTATCGTCAGGGGCTGGGAGT-3′ and 5′-ACTCCCAGCCCCTGACGATACAGTATTGCCTCATAGTTGGTTC-3′ for *SLC1A2* (EAAT2) mut2, 5′-CTTTGCGTAGGCCGTCTGGTTTCTCATGCAGTAAGC-3′ and 5′-TGAGAAACCAGACGGCCTACGCAAA −3′ for *SLC38A2* (SNAT2) mut1, 5′-GCAGTAAGCTTTATAAAAACTCATTTGTGACGGACTGTCATCTCATTCTTGTACAACGTAGA-3′ and 5′-TCTACGTTGTACAAGAATGAGATGACAGTCCGTCACAAATGAGTTTTTATAAAGCTTACTGC-3′ for *SLC38A2* (SNAT2) mut2, 5′-CTACAAACAGGCCTTGTGCTCAACCACCTTAAATGCCTCCTGGG-3′ and 5′-CCCAGGAGGCATTTAAGGTGGTTGAGCACA-3′ for *SLC16A7* (MCT2) mut1, 5′-CCCATTTTCATAACTATGTGAAAACCGTCGGGAATTTGAATCACTCC-3 and 5′-TCAGGAGTGATTCAAATTCCCGACGGTTTTCAC-3′ for *SLC16A7* (MCT2) mut2. The coding sequences of human *SLC1A2* (EAAT2), *SLC38A2* (SNAT2), and *SLC16A7* (MCT2) were synthesized by GeneChem (Shanghai, China) and cloned into the CMV enhancer-MCS-3flag-polyA-EF1A-zsGreen-sv40-puromycin plasmid vector. All constructs were verified by sequencing.

Transfection was performed using Attractene Transfection Reagent (Qiagen, Cat# 301005) following manufacturer's procedures. When indicated, luciferase activities were determined 48 h post-transfection using the Dual-Glo Luciferase Assay System (Promega, Cat# E2920). Cell proliferation was assessed by counting cell numbers every 24 h after seeding 5,000 cells per well in 12-well plates. In general, for cell treatment 2 μg of EVs were added to recipient cells cultured in 2 ml medium in a 6-well plate.

## EV purification and characterization

EVs were purified from conditioned medium (CM) as in our previous studies[37,38]. Cells were cultured in serum-free medium containing 1% BSA for 24 h before CM was collected. The CM was pre-cleared by centrifugation at 500 g for 15 min and then at 10,000 g for 20 min. The supernatant underwent ultracentrifugation at 110,000 g for 70 min and the EV-containing pellet was washed once in PBS under the same ultracentrifugation conditions. When indicated, CFSE (5-(and 6)-Carboxyfluorescein diacetate succinimidyl ester of CFDA SE; BioLegend) or DiI (ThermoFisher Scientific) was added into the PBS at 5 μM and incubated for 1 h at 37 °C before the washing step, followed by an extra round of washing in PBS to remove the excess dye. The EV pellet was suspended in PBS and used in various experiments. Nanoparticle tracking analysis (NTA) was performed using a NanoSight NS300 (Malvern Panalytical). RNA extraction from EVs was carried out using TRIZOL LS (Thermo Fisher Scientific) following manufacturer's protocol. To determine the topology of EV RNA, EVs were treated with Proteinase K (10 μg/ml) followed by RNase If (40 U) in the presence or absence of 1% Triton X-100 prior to RNA extraction and miRNA detection. Iodixanol/OptiPrep density gradient centrifugation was used to fractionate EVs that were enriched by CM ultracentrifugation.

## Brain slice culture

Organotypic brain slice cultures were adapted from previously described protocols[31,33]. Brains from female mice of ~8-week-old were dissected in complete HBSS (HBSS supplemented with 2.5 mM HEPES pH 7.4, 30 mM glucose, 1 mM $CaCl_2$, 1 mM $MgSO_4$, and 4 mM $NaHCO_3$) and immediately embedded in 4% low melting point agarose (Lonza) in plastic embedding molds. After the agarose hardened, the embedded brain and agarose were removed and trimmed prior to sectioning using a VT1000S vibratome (Leica Biosystems, Deer Park, IL). Coronal slices of 300 μm were cut on ice and intact slices were transferred onto the top of polycarbonate cell culture inserts with 0.4-μm pore size (Cat# 140652, ThermoFisher Scientific, Waltham, MA), which were placed in 12-well plates containing DMEM supplemented with 25% complete HBSS, 5% dialyzed FBS (Gibco), 1 mM glutamine, penicillin/streptomycin, and 50 μg/mL Normocin (InvivoGen, San Diego, CA). Damaged slices were discarded. Brain slices were cultured at 37 °C in a humidified incubator with 5% $CO_2$ and media were changed every two days. EVs (10 μg per treatment) were added on the top of brain slices on day 2 and once again on day 3. CM and brain slices were harvested for experiments on day 4. In some experiments, $10^4$ GFP-labeled MDA-231 or MDA-231-BR3 cells were added to the brain slice on day 2, and the

GFP fluorescence was visualized on day 5 using a BZ-X700 fluorescence microscope (Keyence, Osaka, Japan) and quantified by a Varioskan LUX multimode microplate reader (ThermoFisher Scientific) after tissue digestion and cell dissociation using NeuroCult enzymatic dissociation kit for adult mouse/rat CNS tissue (STEMCELL Technologies, Vancouver, Canada). To assess cell viability in brain slices, EV-treated brain slices were dissociated and the cells were assessed using PrestoBlue Cell Viability Reagent (ThermoFisher Scientific) following the manufacturer's protocol.

### Medium metabolite analyses

Levels of glutamate, glutamine, and lactate in CM were measured using corresponding assay kits (Cat# EGLT-100, EGLN-100, and ECLC-100) from Bioassay Systems (Hayward, CA) following manufacturer's instructions. To examine glutamate consumption by NHA, cells were treated with EVs for 48 h, switched to medium containing no glucose, FBS, or glutamate for 24 h, and then switched to glucose/FBS-free medium containing 3 mM glutamate. CM were collected after 24 h and the remaining glutamate levels were measured. To examine glutamine consumption by differentiated SHSY-5Y, cells were treated with EVs for 48 h, switched to medium containing no glucose, 10% dialyzed FBS, and no glutamine for 24 h, and then switched to glucose-free medium containing 10% dialyzed FBS and 2 mM glutamine. CM were collected after 48 h and the remaining glutamine levels were measured. To examine lactate consumption, differentiated SHSY-5Y cells were treated with EVs for 48 h, switched to medium containing no glucose, 10% dialyzed FBS, and no glutamine for 24 h, and then switched to medium containing 1 g/L glucose, 10% dialyzed FBS, and 25 mM lactic acid. CM were collected after 12 h and the remaining lactate levels were measured. In some experiments the CM were diluted with PBS to ensure readings were within detection linearity.

### Animals

All animal experiments were approved by the institutional animal care and use committee (IACUC) at the University of California San Diego and City of Hope Beckman Research Institute. This study is compliant with all relevant ethical regulations regarding animal research. All mice used in this study were obtained from the Jackson Laboratory. Female NOD/SCID/IL2Rγ-null (NSG) mice of ~8-week-old were used in all experiments except for preparation of brain slices, in which female C57BL/6 mice of ~8-week-old were used. Mice were maintained in the City of Hope or University of California San Diego small animal facility under 12-hour light/12-hour dark cycle (on at 6:00 am/off at 6:00 pm) with temperature of 72 °F ± 2 and 30-70% humidity. The maximal tumor size/burden permitted by the institutional animal care and use committee was a single mammary tumor reaching 1.5 cm in diameter. This maximal tumor size/burden was never exceeded in our mouse experiments. During the study, mice were euthanized when one of the followings was observed: failure to eat food or drink water, failure to make normal postural adjustments or display normal behavior, obvious distress, 20% body weight loss (compared to normal), and excessive tumor burden (tumor diameter reaches 1.5 cm).

For EV treatment, EVs from indicated cells were injected into the tail vein twice a week for 5 weeks (~10 μg EVs per injection per mouse). Some mice were sacrificed at this point for collection of the whole brains, which were used for cryosectioning followed by immunofluorescence, tissue homogenization followed by Western blot analysis, or tissue dissociation using the NeuroCult enzymatic dissociation kit for adult mouse/rat CNS tissue. After dissociation, the obtained cell portion was used for RNA extraction and qPCR. The interstitial portion that was separated from the cell portion was used for measurements of glutamate, glutamine, and lactate using the kits described for medium metabolite analyses. Some EV-treated mice subsequently received an injection of $1 \times 10^6$ MDA-MB-231-HM cells mixed with Matrigel (BD Biosciences, Franklin Lakes, NJ) into the No.4 mammary fat pad.

The MDA-MB-231-HM cells were generated through explant culture of a spontaneous meningeal metastasis of MDA-MB-231 cells and have been used in our previous study[48]. Tumor growth was monitored starting from 2 weeks after tumor cell injection, and tumor volume was calculated using the formula (length × width$^2$)/2. In vivo bioluminescent imaging was carried out using a Xenogen system (Caliper Life Sciences, Waltham, MA). Some other EV-treated mice subsequently received an injection of $1 \times 10^6$ T47D-BR2 cells into the left ventricle. As T47D tumor growth in mice is dependent on estrogen, these mice also received weekly subcutaneous injections of 2 mg/kg estradiol cypionate (Cat# S4046; Selleck) following the intracardiac tumor cell injection. The EV treatment continued for three weeks in these mice. Five weeks after the intracardiac injection, in vivo bioluminescent imaging was carried out, followed by collection and bioluminescent imaging of individual organs. Paraffin-embedded brain tissue sections were used for histological examination by hematoxylin and eosin (H&E) staining. Briefly, tissue sections on coated slides were dewaxed with xylene and gradient alcohol after being incubated in an oven at 70 °C for 1 h, counterstained with H&E, dehydrated, and covered.

For intracranial tumor cell injections, after the mouse is anesthetized, a sagittal incision of ~1 cm long was first made over the parieto-occipital bone. Then, a micro drill was used to drill a hole in the skull at 2 mm to the right of the bregma and 1 mm anterior to the coronal suture, creating an opening for the injection of cancer cells to establish intracerebral tumor. To ensure that the appropriate injection depth was achieved, a 3-mm section cut off from the pointed end of a P20 pipetteman tip was fitted over the syringe to limit the injection depth, ensuring that the tip of the syringe needle is 3 mm from the underside of the skull. Luciferase-expressing MDA-231/ctrl or MDA-231/199b cells, or mCherry-expressing BR3/anti-ctrl or BR3/anti-199b cells ($5 \times 10^4$ cells in 3 μL saline) were slowly injected into the hole previously created. The hole was then sealed using sterile bone wax and veterinary skin glue was used to close the skin. Growth of the luciferase-expressing MDA-231/ctrl and MDA-231/199b tumors were monitored by weekly in vivo bioluminescent imaging. Three weeks later, some mice were administered with L-glutamine-$^{13}C_5$ (Sigma-Aldrich) solution via tail-vein infusion following a previously described method[70]. A 27 G catheter was placed in the lateral tail vein under anesthesia. Using a syringe pump (Braintree Scientific; Braintree, MA), L-glutamine-$^{13}C_5$ solution (40 mM) was infused at the speed of 700 μL/h over a total period of 30 min. The mouse was sacrificed 1 h after infusion and the whole brain was removed and dissociated using the NeuroCult enzymatic dissociation kit for adult mouse/rat CNS tissue. The cell portion was saved for measurement of mCherry fluorescence, which was used as a relative quantification of the growth of mCherry-expressing BR3/anti-ctrl and BR3/anti-199b tumors. The cell-free interstitial portion was further centrifuged to remove any remaining cells and debris, and then dried in a vacuum concentrator for subsequent metabolite analyses. The weight of each dried sample was recorded and used for normalization of levels of NMR-determined metabolites.

### Metabolite profiling by NMR spectroscopy

All NMR analyses were performed at the City of Hope NMR Core facility as described previously[70]. Sample preparation and data analysis were performed as described[70]. The dried pellet of brain extracellular extract was resuspended in 500 μL 100% D$_2$O containing 5 mM sodium 2,2-dimethyl-2-silapentane-5-sulfonate (DSS; Cambridge Isotope Laboratories, Tewksbury, MA) that serves as an internal chemical shift reference and a concentration standard in 2D spectra. NMR spectra were acquired at 25 °C on a Bruker Avance spectrometer equipped with a cryoprobe operating at 600.19 MHz $^1$H frequency. Two-dimensional constant-time $^1$H-$^{13}$C HSQC (Heteronuclear Single Quantum Coherence) was used. The spectrum width for $^1$H and $^{13}$C are 16 ppm and 14 ppm, respectively. $^1$H-$^{13}$C correlation spectra were

 

processed using Bruker topspin 3.1, and analyzed with Sparky software (T.D. Goddard and D.G. Kneller, SPARKY 3, University of California, San Francisco). The order for sample preparation and data collection of biological samples were randomized.

## RNA extraction, reverse transcription, and quantitative PCR

These procedures were carried out as reported[48,70]. Sequences of the primers were obtained from PrimerBank and listed in Supplementary Information. For mRNA detection, reverse transcription (RT) was performed using random primers; 18 S rRNA was used as an internal reference in PCR to calculate the relative level of each mRNA. For miRNA detection, TaqMan® miRNA assays were used, with gene-specific RT primer, TaqMan probe, and PCR primer set for each miRNA. The U6 primer was used as an internal reference for intracellular miRNA levels. As a spike-in reference for EV miRNA levels, 10 fmol of synthetic Arabidopsis thaliana miR159a (ath-miR159a) was added during RNA extraction and measured for data normalization. PCR data were collected and analyzed using Bio-Rad CFX Manager software (Bio-Rad, version 3.1).

## Small RNA-seq and data analysis

Illumina sequencing was performed by the City of Hope Integrative Genomics Core using RNA samples extracted from patient sera and from cultured cells and EVs. Samples were independently subjected to library preparation and deep sequencing. All small RNAs of 15–52 nts were selected and sequenced using the Hiseq 2500 system, following the manufacturer's protocol (Illumina, San Diego, CA). Raw counts were normalized by trimmed mean of M value (TMM) method and differentially expressed miRNAs between patients with and without brain metastasis or between different cell lines were identified using Bioconductor package "edgeR". The miRNAs will be regarded as differentially expressed when their P values were <0.05, log2 fold change ≥ 1, and average CPM (in log2 scale) in one group ≥ log2(10).

## Single-cell RNA-seq analysis

Gene Expression Omnibus (GEO) dataset GSE168408 was re-analyzed to determine the expression patterns of selected genes. The "Seurat" package was used for processing single-cell RNA-seq data, including data filtering (cells and genes), normalization, principal component analysis (PCA), and t-distributed stochastic neighbor embedding (t-SNE). For quality control, we removed single cells that had fewer than 300 genes or more than 8000 genes to respectively exclude ruptured cells and potential non-singlet cells. Given the common problem of low viability when single cells are isolated from tissues, we then filtered out cells with more than 10% mitochondrial and more than 3% HB genes to remove dying cells and blood cells. We set the minimum level of unique molecular identifier (UMI) count of each cell at 100,000 to illuminate low-depth data. As such, 49,820 cells and 27,319 genes were included in this analysis. Subsequently, we used Harmony algorithm to integrate and remove batch effect. After cell filtering, the single-cell RNA-seq data of high-quality cells were normalized to find highly variable genes for downstream analyses. Then, PCA was done on highly variable genes to identify significant principal components (PCs). Cell clustering was performed using the top 18 PCs and the t-SNE algorithm. Annotation of cell type in different cell clusters was performed using the "SingleR" package (Bioconductor version 3.11) for automatic cell annotation and five major cell subsets were obtained.

## Western blot analysis

Cells were lysed in RIPA Lysis and Extraction Buffer (Thermo Scientific, Catalog # 89901) supplemented with cOmplete™ Protease Inhibitor Cocktail (Roche, Cat# 04693124001) and Halt™ Phosphatase Inhibitor Single-Use Cocktail (Thermo Scientific, Cat# 78428). Tissues were homogenized in protease and phosphatase inhibitor-supplemented RIPA buffer using a Precellys 24 tissue homogenizer (Bertin

Technologies). Protein concentrations in cell and tissue lysates were measured using the Pierce™ BCA Protein Assay Kit (Thermo Scientific, Cat# 23225). An equal amount of protein extracts (~30 μg) were subjected to electrophoresis on a 10% or 12% SDS polyacrylamide gel and then transferred onto a PVDF membrane (Bio-Rad, Cat# 1620177). All antibodies used in this study are listed in Supplementary Information. Horseradish peroxidase-conjugated secondary antibodies were used for all Western blots. Signals were detected using Pierce™ ECL Western Blotting Substrate (Thermo Scientific, Cat# 32106) or SuperSignal™ ELISA Pico Chemiluminescent Substrate (Thermo Scientific, Cat# 37069). Unprocessed original scans of immunoblots are available in source data.

## Immunofluorescence

Following a previously described procedure[48], O.C.T. sections and brain slices were fixed with 4% PFA in PBS, blocked and permeabilized with PBS containing 10% goat serum and 0.05% saponin, prior to incubation with anti-GFAP diluted 1:500 (Abcam, Cat# ab7260) for astrocytes or anti-mouse MAP2 diluted 1:100 (Abcam, Cat# ab27957 or ab5392) for neurons, in some cases together with mouse anti-human CD63 diluted 1:50 (MEM-259, Cat# NB100-77913, Novus Biologicals, Littleton, CO). Secondary antibodies included anti-rabbit Alexa 647 or Alexa 488 diluted 1:300 (Cat# A-21244 or A-11070, ThermoFisher Scientific), anti-mouse Alexa 594 diluted 1:300 (Cat# A-11032, Thermo-Fisher Scientific), and anti-chicken Alexa 647 (Cat# A-21449, ThermoFisher Scientific). Slides were then stained with Nuclei were stained by DAPI (4′, 6-diamidino-2-phenylindole). Images of brain slices were captured using a STELLARIS 8 DIVE 2-photon functionality on a DM8 upright microscope (Leica Microsystems, Wetzlar, Germany) and processed using LAS X 4.4 software (Leica Microsystems). Other images were captured using a ZEISS LSM 880 confocal microscope system (Carl Zeiss, Oberkochen, Germany) and processed using Image Pro Premier 9.3 software (Media Cybernetics, Rockville, MD).

## Human serum specimens

Archived samples from cancer patients used in this study were collected in accordance with the Declaration of Helsinki and the principles of Good Clinical Practice. All participants provided written informed consent. Human serum specimens were obtained from voluntarily consenting breast cancer patients between February 2006 and December 2011 at the City of Hope National Medical Center (Duarte, CA, USA) under institutional review board-approved protocols. The study did not involve classifications based on race, ethnicity, or other socially relevant groupings. The genetic ancestry data were not available to researchers. All 42 patients involved in this study were females with stage IV disease either with or without brain metastases at the time metastatic disease was diagnosed. Among them, 21 patients had brain metastases, in some cases with concurrent metastases to other organs, whereas the other 21 patients had distant metastases to other organs without the involvement of central nervous system. The two groups exhibited balanced age, tumor subtype, and sample collection time. Serum specimens examined in this study were collected at the time metastasis was initially diagnosed or the earliest draw available. Clinical characteristics are summarized in Supplementary Information. Trizol LS (ThermoFisher Scientific) was used to extract total RNA from ~0.5 mL of serum; RNA pellet was dissolved in 10 μL of RNase-free water and subjected to small RNA-seq and RT-qPCR.

## Statistics and reproducibility

All quantitative data are presented as mean ± standard deviation (SD). Statistical significance was calculated using Prism (GraphPad Software, version 7.01). Two-tailed Student's t-tests were used for comparison of means of data between two groups. For multiple independent groups, one-way or two-way ANOVA with Tukey's tests were used. Values of $P < 0.05$ were considered significant. Sample size was generally chosen

based on preliminary data indicating the variance within each group and the differences between groups. Except for small RNA-seq, all experiments were performed at least twice independently with similar results. No samples or animals were excluded from the analysis. All mice/samples were randomized before experiments. Data collection and analysis were performed blinded to group allocation. For studies with cell cultures, biological replicates are referred to as independent dishes of cells receiving the same treatment that were processed on the same days or on different days.

## Reporting summary

Further information on research design is available in the Nature Portfolio Reporting Summary linked to this article.

## Data availability

Source data are provided with this paper. Small RNA-seq data that support the findings of this study are deposited in the Gene Expression Omnibus (GEO) under accession code GSE216934. Dataset GSE168408 is re-analyzed in this study. The NMR data generated in this study have been deposited in the Zenodo database under accession code https://doi.org/10.5281/zenodo.10927179 All the other data are available within the article and its Supplementary Information.

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

## Author contributions

S.E.W. conceived ideas. N.C.S. and R.F.H. contributed to project planning. X.R. and S.E.W. designed and performed most of the experiments. W.Y., M.C., M.Y.F., and J.W. assisted with mouse experiments and cell line construction. X.L. assisted with PCR experiments. R.A.M.D. and R.F.H. assisted with brain slice preparation. K.Y. assisted with data analyses. M.P. assisted with clinical sample assembly. R.J. provided the BBM models. W.Y. and Y.C. assisted with NMR analyses. X.W. and A.L. assisted with small RNA-seq and bioinformatics. S.E.W. and X.R. wrote the manuscript.

## Competing interests

The authors declare no competing interests.
