## [Peer Review File · Nature Communications]

Breast Cancer Cell-secreted miR-199b-5p Hijacks Neurometabolic Coupling to Promote Brain MetastasisREVIEWER COMMENTS

Reviewer #1 (Remarks to the Author):

This manuscript presents a fascinating hypothesis, that brain metastasizing tumor cells hijack astrocyte-neuron metabolic communication. If true, this will be an outstanding addition to the literature. A number of technical issues are listed below that require thorough examination before this conclusion can be verified.

Overarching issues listed include the use of one high miR199b expressing cell line. We have no idea of how levels of this miR affect the biology. Also, empty EVs at times have intermediate effects- a better control would be to also test EVs with an unrelated miR. The brain metastasis assays are substandard. Images showing putative expression patterns cannot be clearly seen.

1. Can the analysis be repeated with samples that will also discriminate lung and liver metastases in Fig. 1?
2. Do other miRs bind to this sequence shown on Fig. 3?
3. Are SLC1A2 only expressed by astrocytes and not neurons? Are SLC38A2 and SLC16A7 only expressed by neurons and not astrocytes? How about expression in other brain cell types?
4. The experiments shown on Fig. 4E are critical. The empty EVs exerted some effect. Pls add an EV with another miR.
5. The differences in Gln for neurons in 4E may be statistically significant but are truly minimal. Do the authors have any proof that they are of functional significance? How much miR-199b is used? Do EVs with different levels of this miR have graded effects?
6. Can the experiments on Fig. 5 be replicated using normal versus brain metastatic tumor cells?
7. On Fig. 5B this reviewer cannot clearly see uptake. I was expecting a yellow signal. I also cannot tell what cells are red.
8. Can the experiments on 5E-F be repeated to include a non-brain metastatic line?
9. The images in Fig. 6A are uninterpretable. Why was the regular 231 line used?
10. An effect of control EVs is seen again in Fig. 6E. This experiment needs another, unrelated miR tested.
11. An effect on metastases in 6F needs to also show whole body imaging and brain histology in H&E stained sections. Two model systems are appropriate for a fundamental observation.
12. The experiment on Fig. 7 needs to be reperformed using a hematogenous brain metastasis assay. This is not metastasis, its intracerebral growth.

Reviewer #2 (Remarks to the Author):

In this manuscript, Ruan et al. report that extracellular vesicle (EV)-carried miR-199b promotes breast cancer brain metastasis. The authors show that miR-199b suppresses astrocyte EAAT2 and neuron SNAT2 and MCT2, leading to the accumulation of glutamine and lactate, which promotes metastasis growth. This is an interesting mechanism, but the conclusions need to be strengthened.

Specific points:

1. The conclusions heavily rely on the MDA-MB-231 cell line and its subline. To strengthen the conclusion, the authors should use additional cancer cell lines to demonstrate that EV miR-199b promotes brain metastasis in vivo.
2. Glutamine levels are relatively high in the blood and brain. The authors show that EV miR-199b significantly changed glutamine concentration in the brain. How about other common sites of breast cancer metastasis (the lung and bone)?
3. Will the administration of glutamine and lactate in mice promote breast cancer brain metastasis in vivo? How about lung and bone metastasis?
4. Will silencing miR-199b (in breast cancer cells) or its extracellular form block brain metastasis in vivo?

Reviewer #3 (Remarks to the Author):

In their manuscript, Ruan and colleagues propose a mechanism by which metastatic Breast Cancer cells alter the metabolic coupling between neurons and astrocytes in the brain to promote the growth of brain metastases. This is an interesting and important question; however, there are several major methodological problems that need to be addressed:

The authors suggest that breast cancer cells release extracellular vesicles (EVs) that contain high levels of miR-199b, which is found at higher levels in the blood of Breast Cancer patients with brain metastases compared to those with metastases in other organs, which is in line with previous work that demonstrated that hsa-mir-199b is uniquely up-regulated in metastatic brain tumors, compared to primary brain tumors results (e.g., PMID: 32906592; table 1).

Subsequently, the researchers propose that miR-199b-5p targets specific solute carrier transporters in the brain; however, the reasoning leading to this hypothesis is unclear. Additionally, the authors state that the analysis of microRNA.org demonstrated an association of miR-199b-5p with EAAT2/GLT1, SNAT2, and MCT2. Nevertheless, an analysis using DIANA tools (TarBase V.8) or rncentral.org did not support this association. Moreover, the authors did not address the regulation of EAAT1 (SLC1A3/GLAST), which is the second main glutamate transporter in astrocytes, by miR-199-5p. Interestingly, DIANA/rncentral.org analyses suggested that miR-199-5p is linked to the expression of the SLC1A5/ASCT2 transporter, which was suggested to play a role in glutamate reuptake by astrocytes (reviewed in PMID: 30234109). These unresolved issues require further attention. Furthermore, the changes in the rLuc reporter levels (Figure 3) appear minimal, despite being statistically significant, according to the authors. It remains uncertain whether these mutations affected glutamate influx into astrocytes or influxes of glutamine and lactate into neurons. Therefore, the functional effect of these mutations, as with the mimic miR, should be determined.

Next, the authors examined the impact of miR-199b encoded EVs on Glu/Gln/Lac uptake and release (Figure 4). Initially, they tested whether NHA or differentiated SH-SY5Y cells took up the fluorescently labeled EVs, but their analysis only suggests a possible association between the EVs and cells. To obtain more accurate results, a z-stack confocal-based analysis with membranous staining, as demonstrated in PMID: 35225262, should be conducted (this should also be performed for 5B). Subsequently, in Figures 4B and 4C, the authors attempted to regulate the levels of EAAT2, SNAT2, and MCT2 by overexpressing or inhibiting miR-199b. Although the mRNA data showed statistical significance, the manipulation of miR-199b resulted in only minor changes in the transcript levels of EAAT2, SNAT2, and MCT2. The immunoblot analyses seemed more promising, but it is difficult to evaluate them due to the overexposure of the housekeeping protein GAPDH and the lack of statistical data on different biological repeats. Additionally, the statistical analysis of the metabolic measurements in Figure 4E should be redone, particularly with regard to the Gln analyses and the reported $P < 0.001$.

Importantly, these experiments lack a clear causal relationship between the observed changes in transcript/protein levels and the metabolic changes. This should be addressed by, for example, knocking down EAAT2 in the cells and demonstrating that miR-199b-based manipulation has no effect on Glu uptake levels in those cells.

The measurement of mCherry fluorescent intensity in Figures 5E and 5F does not necessarily correlate with "Enhanced growth of cancer cells." The observed results could be associated with the regulation of mCherry expression itself, cell proliferation, cell cycle arrest, or cell death. To support their hypothesis and obtain more accurate results, the authors should conduct a non-transcriptionally regulated experiment, such as CFSE staining. Of note, using siRNA to support the author's hypothesis (Figure 5F) is a step in the right direction towards establishing a clear causal relationship, but they lack key controls demonstrating the target transcript/protein was indeed downregulated and identifying the cells in which it was downregulated.

The authors should clarify what is the percentage of astrocytes and neurons that up took the EVs, those are found in proximity to the tumor cells. Is the use of anti-CD63 mAbs the best way to analyse uptake of EVs? Wouldn't it be degraded in the cells that up took the EVs? The results in Panels B-E are quite unexpected compared to the minimal regulation demonstrated in the in-vitro

experiments, and the low "uptake" of the EVs shown in panel A. more over it is clear that those changes are happening in astrocyte or neurons. According to the immunoblot data in panel 6D (that should be revised, similar to previous comments on the authors immunoblots), the authors should perform immunostaining coupled to GFAP and MAP2 staining in sham and tumor-bearing mice.

In-vivo experiments (Figure 6) - The authors should clarify the percentage of astrocytes and neurons that took up the EVs in proximity to the tumor cells. Additionally, it should be noted whether using anti-CD63 monoclonal antibodies is the most effective way to analyze EV uptake, as it is possible that CD63 may degrade in the cells that have taken up the EVs. The results in Panels B-E are unexpected compared to the minimal regulation demonstrated in the in-vitro experiments and the low "uptake" of EVs shown in Panel A. Moreover, it is unclear whether these changes occur in astrocytes or neurons. However, as the immunoblot data in Panel 6D, which should be revised as per previous comments on the authors' immunoblots, seems to suggest dramatic changes in the protein expression, the authors should conduct immunostaining coupled with GFAP and MAP2 staining in sham and tumor-bearing mice to address this question.

It is worth noting that miR-199b has been shown to regulate cancer progression in several cancer types through various mechanisms. However, the current in vivo experiments demonstrate a correlation between miR-199b-carrying EVs, metabolic pathways involving Glu/Gln/Lac, and the astrocytes-neurons axis and tumor cells. There is still no clear causal relationship. This issue should be experimentally addressed, or alternatively, the limitations of the in vivo (and ex-vivo) data should be clearly discussed in the results, discussion, and abstract sections.

Additional comments:

1. The authors ought to present a more comprehensive background on the involvement of astrocytes in primary and secondary brain tumors in the introduction or discussion section. Specifically, they should explore the role of astrocytes in the metabolic reprogramming of the tumor microenvironment (TME). This could be achieved by referencing relevant studies such as PMID 29892069, 35899587, 27225120, and 25639230.
2. It would be beneficial for the authors to provide a more extensive context regarding the role of miR-199b in cancer generally, as well as in relation to breast cancer and metastasis. This could be accomplished by citing relevant studies such as PMID 23296799, 35090460, and 32082362, among others.
3. The authors need to be more consistent and clearer about the identity of miR-199b, as miR-199b-5p has a different sequence from miR-199b-3p. They should ensure that they are unambiguously referring to miR-199b-5p throughout the manuscript. Doing so will help avoid any confusion and inaccuracies in interpreting their results.
4. It would be advantageous to provide one or two lines of summary or conclusion for the section titled "miR-199b is associated with brain metastasis in BC patients and secreted by brain-tropic MBC cells".
5. The authors should incorporate an immunoblot analysis of CD63 in the BBM1 extracellular vesicles (EV) shown in Figure 2B.

Reviewer #4 (Remarks to the Author):

In this study, the authors demonstrated that EVs derived from brain-metastatic breast cancer cells shuttled miR-199b-5p to neurons and astrocytes, affecting neural metabolism and promoting brain metastases. Their work is great. I guess that experiments in this study needed very tough work. This manuscript is worth considering for publication in this journal, but it can be considered after addressing the following issues in revision.

Major comment

1. It is interesting that breast cancer cell-derived EVs affect neural metabolism, leading to brain metastases. However, as the authors know, this journal has a high impact on this research field, please emphasize how this idea is novel. They should develop the topic in the discussion section.
2. In the introduction section, please discuss the relationship between neural metabolites and brain

(metastatic) tumor in more detail. Please describe how critical the level of glutamate, glutamine, and lactate for tumor growth.

3. Please make figure 5(A) understandable. In the experiments using brain slices, it is slightly difficult to understand the time course of the experiments. In some experiments, cancer cells were seeded on the same day (day 2) with EV treatment, but current figure 5(A) doesn't show this time course correctly. Moreover, in figure 5(A), the illustrations of EV treatment and cancer seeding are the same. It is confusing. Please change the illustration.

4. In the experiments using brain slices, they demonstrated that the levels of glutamine and lactate were increased by EV treatment to brain slices, leading to significant cancer cell growth. These data are so amazing. They need to carefully verify the validity of these results. Please check the following points. First, EVs could be also taken up by cancer cells on the brain slice. They need to show data on EV-treated cancer cell growth. They showed data on MDA-231/miR-199b in vitro cell growth in figure 7(C). According to this data, I guess that cancer cell growth is down-regulated by EV treatment, but they need to show clearly. Second, they need to confirm whether the cancer cell growth is changed when the levels of glutamine and lactate in CM are up or downregulated. Thirdly, living cell numbers in the brain slices could be reduced by EV treatment. Current results could be obtained when EV treatment causes cell death and reduce living cell numbers. Finally, SLC transporters can be expressed not only in astrocytes and neurons but also in other types of cells in the brain slices. Please examine and discuss these points.

5. The authors need to show the bioluminescent image of the body in in vivo experiments (Figure 6). I am interested in the specificity of EV-miR-199b to the brain metastatic tumor. I wonder if EV-miR-199b have some effect on the tumor in other sites such as lung and lymph nodes. If they confirm that EV-miR-199b does not have any effect on lung or lymph nodes, they succeed in demonstrating that EV-miR199b function is specific to brain.

6. The authors should discuss the relationship between BBB and cancer cell-derived EV microRNAs more in detail. Please cite a following paper. Moreover, does miR-199b-5p have some function on BBB? They should confirm at least the difference of BBB penetration efficiencies between MDA-231/miR-199b EV and MDA-231/ctrl EV (Figure 6A).

Tominaga, N., Kosaka, N., Ono, M. et al. Brain metastatic cancer cells release microRNA-181c-containing extracellular vesicles capable of destructing blood-brain barrier. *Nat Commun* 6, 6716 (2015). <https://doi.org/10.1038/ncomms7716>

Minor comment

0. Please show how to label cancer cells by mCherry in the methods section.

1. Please show the scale bar on the left picture in figure 5(B).

Response to Reviewers

We thank the editor and all reviewers for the constructive comments. In response to these comments, we made substantial revisions to the manuscript and added a considerable amount of new data from *in vitro* and *in vivo* experiments. Due to multiple factors that were out of our control in the past year, the revision process took much longer than usual. We appreciate your kind understanding. Below please find our point-by-point responses.

Reviewer #1:

This manuscript presents a fascinating hypothesis, that brain metastasizing tumor cells hijack astrocyte-neuron metabolic communication. If true, this will be an outstanding addition to the literature. A number of technical issues are listed below that require thorough examination before this conclusion can be verified. Overarching issues listed include the use of one high miR199b expressing cell line. We have no idea of how levels of this miR affect the biology. Also, empty EVs at times have intermediate effects- a better control would be to also test EVs with an unrelated miR. The brain metastasis assays are substandard. Images showing putative expression patterns cannot be clearly seen.

Response: We highly appreciate these constructive comments and have generated a new T47D cell line model with miR-199b overexpression as well as a new cell line secreting EVs carrying an unrelated miRNA (miR-211) as an additional control. The new results are summarized below in our responses to individual comments and have been added to the revised manuscript.

1. Can the analysis be repeated with samples that will also discriminate lung and liver metastases in Fig. 1?

Response: We were able to obtain a lung-metastasizing variant of MDA-MB-231 (MDA-231-LM2) and a variant of T47D derived from a spontaneous brain metastasis (T47D-BR2). The levels of miR-199b in these additional cell line models and their EVs were measured by qPCR and results are added to new Fig. 1d. Unfortunately, for the clinical samples assessed in Fig. 1, we do not have the complete clinical information regarding the status of lung and liver metastases.

2. Do other miRs bind to this sequence shown on Fig. 3?

Response: The sequences shown in Fig. 3 (current Fig. 2a) are not predicted as binding sites for other miRNAs.

3. Are SLC1A2 only expressed by astrocytes and not neurons? Are SLC38A2 and SLC16A7 only expressed by neurons and not astrocytes? How about expression in other brain cell types?

Response: To obtain a comprehensive understanding of these genes' expression patterns in different types of brain cells, we re-analyzed a single cell gene expression dataset of human prefrontal cortex samples (GSE168408). Following cell clustering and cell type annotation, five major cell subsets were obtained (new Supplementary Fig. 3a). Expression of EAAT2 (SLC1A2) was the highest in astrocytes, whereas expression of SNAT2 (SLC38A2) and MCT2 (SLC16A7) were the highest in excitatory neurons (new Supplementary Fig. 3b,c). Oligodendrocytes also show some EAAT2 and MCT2 expression; however, in this study we focus on astrocytes and neurons for their metabolic coupling pathway.

4. The experiments shown on Fig. 4E are critical. The empty EVs exerted some effect. Pls add an EV with another miR.

Response: We created a new MDA-MB-231-derived cell line that overexpressed an unrelated miRNA (miR-211) and secreted EVs carrying a high level of miR-211 as an additional control. This unrelated control miRNA did not alter the expression of SLC genes or nutrient consumptions, as shown in the new data added to new Fig. 3b and 4a.

5. The differences in Gln for neurons in 4E may be statistically significant but are truly minimal. Do the authors have any proof that they are of functional significance? How much miR-199b is used? Do EVs with different levels of this miR have graded effects?

Response: We repeated the experiments in Fig. 4E and tested two doses of high-miR-199b EVs in the treatment. The effect of high-miR-199b EVs on nutrient consumptions exhibited a dose-dependent pattern, as shown in the new data added to new Fig. 4a. In addition, we also overexpressed EAAT2, SNAT2, or MCT2 using plasmids carrying the cDNA (without 3'UTR) of each target gene, and showed that restoration of gene expression blocked the effect of miR-199b-5p (new Fig. 4b-d).

6. Can the experiments on Fig. 5 be replicated using normal versus brain metastatic tumor cells?

Response: We repeated the experiments by testing the growth of both parental MDA-MB-231 and the brain-metastasizing MDA-231-BR3 cells on EV-treated brain slices. The new results are added to new Fig. 5e,f.

7. On Fig. 5B this reviewer cannot clearly see uptake. I was expecting a yellow signal. I also cannot tell what cells are red.

Response: We have improved the resolution of the images. GFAP is used as an astrocyte marker and is localized in cytoplasmic intermediate filaments, whereas MAP2, a neuron marker, is mainly localized to the plasma membrane and microtubules in the cytosol. Therefore, we do not expect significant overlay between these cell type markers and EV signals (the yellow signals), but rather their existence in the same cell (indicated by a yellow arrow).

8. Can the experiments on 5E-F be repeated to include a non-brain metastatic line?

Response: We repeated the experiments by adding the parental MDA-MB-231 cells. The new results are added to new Fig. 5e,f.

9. The images in Fig. 6A are uninterpretable. Why was the regular 231 line used?

Response: We repeated the immunostaining and fluorescent microscopy using fresh brain tissues from new experiments. The results showing markers of injected EVs and brain cell types (astrocytes and neurons) from mice that had received MDA-231/ctrl or MDA-231/199b EVs are included.

10. An effect of control EVs is seen again in Fig. 6E. This experiment needs another, unrelated miR tested.

Response: We included EVs from MDA-MB-231/miR-211 carrying a high level of miR-211 as an additional control. This unrelated control miRNA did not alter the expression of miR-199b target genes or nutrient levels in the brain, as shown in the new data added to Fig. 6b-e.

11. An effect on metastases in 6F needs to also show whole body imaging and brain histology in H&E stained sections. Two model systems are appropriate for a fundamental observation.

Response: In the spontaneous metastasis model in Fig. 6F, because the luciferase-labeled cancer cells were injected into the mammary fat pad, the bioluminescent signals from the primary tumors were too intense to allow signals in the brain to be seen and we therefore covered the bodies of mice to focus on the brains instead of showing whole body imaging. To overcome this shortage and by following the reviewer's suggestion, we generated a new T47D cell line model with miR-199b overexpression and secretion (new Fig. 1d). In a new mouse experiment we first treated mice with i.v. injected EVs, and then performed intracardiac injection using the T47D cell line to assess the effect of EV miR-199b on brain metastasis. The new results showing whole body imaging are added to new Fig. 7g-i and Supplementary Fig. 6.

12. The experiment on Fig. 7 needs to be reformed using a hematogenous brain metastasis assay. This is not metastasis, its intracerebral growth.

Response: We generated a new T47D cell line model with miR-199b overexpression and secretion (new Fig. 1d). In a new mouse experiment we first treated mice with i.v. injected EVs, and then performed intracardiac injection using the T47D cell line to assess the effect of EV miR-199b on brain metastasis. The new results

showing whole body imaging are added to new Fig. 7g-i and Supplementary Fig. 6. In this model, we observed specifically enhanced metastasis to the brain but not to lung and liver. We also observed metastasis to the ovaries, which could be related to ER signaling in this ER+ tumor model that requires subcutaneous administration of estrogen (estradiol cypionate).

Reviewer #2:

In this manuscript, Ruan et al. report that extracellular vesicle (EV)-carried miR-199b promotes breast cancer brain metastasis. The authors show that miR-199b suppresses astrocyte EAAT2 and neuron SNAT2 and MCT2, leading to the accumulation of glutamine and lactate, which promotes metastasis growth. This is an interesting mechanism, but the conclusions need to be strengthened.

Specific points:

1. The conclusions heavily rely on the MDA-MB-231 cell line and its subline. To strengthen the conclusion, the authors should use additional cancer cell lines to demonstrate that EV miR-199b promotes brain metastasis in vivo.

Response: As suggested by the reviewer, we generated a new T47D cell line model with miR-199b overexpression and secretion (new Fig. 1d). In a new mouse experiment we first treated mice with i.v. injected EVs, and then performed intracardiac injection using the T47D cell line to assess the effect of EV miR-199b on brain metastasis. The new results showing whole body imaging are added to new Fig. 7g-i and Supplementary Fig. 6. In this model, we observed specifically enhanced metastasis to the brain but not to lung and liver. We also observed metastasis to the ovaries, which could be related to ER signaling in this ER+ tumor model that requires subcutaneous administration of estrogen (estradiol cypionate).

2. Glutamine levels are relatively high in the blood and brain. The authors show that EV miR-199b significantly changed glutamine concentration in the brain. How about other common sites of breast cancer metastasis (the lung and bone)?

Response: We analyzed and compared interstitial glutamine and lactate levels in the lung tissues from mice that had received EVs carrying different miR-199b levels. The results showing no difference in lung glutamine/lactate levels have been added to new Fig. 6f. Interstitial glutamate was undetectable in the lung tissues we collected from this experiment.

3. Will the administration of glutamine and lactate in mice promote breast cancer brain metastasis in vivo? How about lung and bone metastasis?

Response: Previous studies have revealed complex tumor regulatory effects of glutamine and lactate that are independent of their function to fuel cancer cell metabolism. In some tumor models, dietary glutamine supplementation inhibits tumor progression through suppressing epigenetically-activated oncogenic pathways, whereas glutamine restriction promotes tumor growth. Lactate in the tumor microenvironment has been shown to promote tumor growth and progression through metabolic and functional regulations of various immune cell subsets. We have added these relevant studies to Discussion.

4. Will silencing miR-199b (in breast cancer cells) or its extracellular form block brain metastasis in vivo?

Response: We generated MDA-231/BR3 cells stably expressing anti-miR-199b to block the function of miR-199b. These cells exhibited decreased ability to grow in the brain and led to lower levels of ¹³C-labeled glutamine and glutamate in the cell-free interstitial fraction in the brain tissue (Fig. 7e,f).

Reviewer #3:

In their manuscript, Ruan and colleagues propose a mechanism by which metastatic Breast Cancer cells alter

the metabolic coupling between neurons and astrocytes in the brain to promote the growth of brain metastases. This is an interesting and important question; however, that are several major methodological problems that needs to be addressed:

The authors suggest that breast cancer cells release extracellular vesicles (EVs) that contain high levels of miR-199b, which is found at higher levels in the blood of Breast Cancer patients with brain metastases compared to those with metastases in other organs, which is in line with previous that demonstrated the hsa-mir-199b is uniquely up-regulated in metastatic brain tumors, compared to primary brain tumors results (e.g., PMID: 32906592; table 1). Subsequently, the researchers propose that miR-199b-5p targets specific solute carrier transporters in the brain; however, the reasoning leading to this hypothesis is unclear. Additionally, the authors state that the analysis of microRNA.org demonstrated an association of miR-199b-5p with EAAT2/GLT1, SNAT2, and MCT2. Nevertheless, an analysis using DIANA tools (TarBase V.8) or rnacentral.org did not support this association. Moreover, the authors did not address the regulation of EAAT1 (SLC1A3/GLAST), which is the second main glutamate transporter in astrocytes, by miR-199-5p. Interestingly, DIANA/rnacentral.org analyses suggested that miR-199-5p is linked to the expression of the SLC1A5/ASCT2 transporter, which was suggested to play a role in glutamate reuptake by astrocytes (reviewed in PMID: 30234109). These unresolved issues require further attention. Furthermore, the changes in the rLuc reporter levels (Figure 3) appear minimal, despite being statistically significant, according to the authors. It remains uncertain whether these mutations affected glutamate influx into astrocytes or influxes of glutamine and lactate into neurons. Therefore, the functional effect of these mutations, as with the mimic mir, should be determined.

Response: We have added the important reference PMID 32906592 pointed out by the reviewer. The three miR-199b target genes identified here (EAAT2, SNAT2, and MCT2) don't contain canonical seed sites for miR-199b, therefore they were not predicted by algorithms requiring seed matching sites. TarBase is a repository for previously reported targets. Since these targets are novel and haven't been previously reported, they are not in TarBase. We also validated these targets by luciferase reporter assay (Fig. 2a) and by RT-qPCR and Western blots upon miR-199b-5p mimic transfection (Fig. 2b,c). To further strengthen the functional effect of miR-199b-5p mediated regulation of EAAT2, SNAT2, and MCT2, we overexpressed EAAT2, SNAT2, and MCT2 using plasmids carrying the cDNA (without 3'UTR) of each target gene, and showed that restoration of these targets blocked the effect of miR-199b-5p (Fig. 4b-d). We also assessed the expression of EAAT1 (SLC1A3/GLAST) and SLC1A5/ASCT2 in astrocytes, and found that EV miR-199b did not alter the expression of these genes. The new results are included in Supplementary Fig. 2c.

Next, the authors examined the impact of mir-199b encoded EVs on Glu/Gln/Lac uptake and release (Figure 4). Initially, they tested whether NHA or differentiated SH-SY5Y cells took up the fluorescently labeled EVs, but their analysis only suggests a possible association between the EVs and cells. To obtain more accurate results, a z-stack confocal-based analysis with membranal staining, as demonstrated in PMID: 35225262, should be conducted (this should also be performed for 5B). Subsequently, in Figures 4B and 4C, the authors attempted to regulate the levels of EAA2, SNAT2, and MCT2 by overexpressing or inhibiting mir-199b. Although the mRNA data showed statistical significance, the manipulation of mir-199b resulted in only minor changes in the transcript levels of EAA2, SNAT2, and MCT2. The immunoblot analyses seemed more promising, but it is difficult to evaluate them due to the overexposure of the housekeeping protein GAPDH and the lack of statistical data on different biological repeats. Additionally, the statistical analysis of the metabolic measurements in Figure 4E should be redone, particularly with regard to the Gln analyses and the reported $P < 0.001$.

Response: We have added z-stack confocal images showing intracellular localization of EVs in NHA and differentiated SH-SY5Y cells. We have also repeated Western blot and metabolite assays; new data are shown in Fig. 3 and Fig. 4.

Importantly, these experiments lack a clear causal relationship between the observed changes in transcript/protein levels and the metabolic changes. This should be addressed by, for example, knocking down EAA2 in the cells and demonstrating that mir-199b-based manipulation has no effect on Glu uptake levels in those cells.

Response: We overexpressed EAAT2, SNAT2, or MCT2 using plasmids carrying the cDNA (without 3'UTR) of

each target gene, and showed that restoration of gene expression blocked the effect of miR-199b-5p (new Fig. 4b-d).

The measurement of mCherry fluorescent intensity in Figures 5E and 5F does not necessarily correlate with "Enhanced growth of cancer cells." The observed results could be associated with the regulation of mCherry expression itself, cell proliferation, cell cycle arrest, or cell death. To support their hypothesis and obtain more accurate results, the authors should conduct a non-transcriptionally regulated experiment, such as CFSE staining. Of note, using siRNA to support the author's hypothesis (Figure 5F) is a step in the right direction towards establishing a clear causal relationship, but they lack key controls demonstrating the target transcript/protein was indeed downregulated and identifying the cells in which it was downregulated.

Response: We chose to use mCherry fluorescence instead of a cell-tracing dye because cell proliferation would result in dye dilution, and CFSE has been used in commercial kits to monitor generations of cell proliferation by dye dilution. Therefore, we cannot use the total CFSE fluorescence intensity as a surrogate for assessment of cancer cell growth. We have repeated the experiments in Fig. 5E,F using MDA-MB-231 and MDA-231-BR3 cells stably expressing GFP, and the new results are included in Fig. 5e,f. To strengthen this result, we also treated cancer cells with the conditioned medium (CM) collected from EV-treated brain slices, with or without supplementation with glutamine or/and lactate, and evaluated cancer cell growth in the CM by quantifying cell number. CM from brain slices treated with high-miR-199b EVs, which contained higher levels of glutamine and lactate, resulted in enhanced cancer cell growth compared to CM from brain slices treated with control EVs. Supplementation of the CM from control EV-treated brain slices with glutamine or lactate to match the levels detected in the CM from high-miR-199-treated groups partially rescue cancer cell growth, whereas adding both glutamine and lactate fully recapitulated the effects of high-miR-199b EVs (new Fig. 5g). In addition, we added Western blot to show the knockdown efficiency of siRNAs (new Supplementary Fig. 4b).

The authors should clarify what is the percentage of astrocytes and neurons that up took the EVs, those are found in proximity to the tumor cells. Is the use of anti-CD63 mAbs is the best way to analyse uptake of EVs? Wouldn't it be degraded in the cells the uptook the EVs? The results in Panels B-E are quite unexpected compared to the minimal regulation demonstrated in the in-vitro experiments, and the low "uptake" of the EVs shown in panel A. more over it is clear that those changes are happening in astrocyte or neurons. According to the immunoblot data in panel 6D (that should be revised, similar to previous comments on the authors immunoblots), the authors should perform immunostaining couple to GFAP and MAP2 staining in shame and tumor-bearing mice.

Response: We performed new immunostaining and fluorescent microscopy using fresh brain tissues from new experiments. The results showing markers of injected EVs and brain cell types (astrocytes and neurons) from mice are included. We have replaced the Western blot images in Fig. 6d. We also added discussion on the limitation of human CD63 detection.

In-vivo experiments (Figure 6) - The authors should clarify the percentage of astrocytes and neurons that took up the EVs in proximity to the tumor cells. Additionally, it should be noted whether using anti-CD63 monoclonal antibodies is the most effective way to analyze EV uptake, as it is possible that CD63 may degrade in the cells that have taken up the EVs. The results in Panels B-E are unexpected compared to the minimal regulation demonstrated in the in-vitro experiments and the low "uptake" of EVs shown in Panel A. Moreover, it is unclear whether these changes occur in astrocytes or neurons. However, as the immunoblot data in Panel 6D, which should be revised as per previous comments on the authors' immunoblots, seems to suggest dramatic changes in the protein expression, the authors should conduct immunostaining coupled with GFAP and MAP2 staining in sham and tumor-bearing mice to address this question. It is worth noting that mir-199b has been shown to regulate cancer progression in several cancer types through various mechanisms. However, the current in vivo experiments demonstrates a correlation between mir-199b-carrying EVs, metabolic pathways involving Glu/Gln/Lac, and the astrocytes-neurons axis and tumor cells. There is still no clear causal relationship. This issue should be experimentally addressed, or alternatively, the limitations of the in vivo (and ex-vivo) data should be clearly discussed in the results, discussion, and abstract sections.

Response: We performed new immunostaining and fluorescent microscopy using fresh brain tissues from new experiments. The results showing markers of injected EVs and brain cell types (astrocytes and neurons) from

mice are included. The percentages of astrocytes and neurons that took up the EVs have been added (Fig. 6a and Supplementary Fig. 5). We were unable to obtain images of endogenous brain metastasis-derived EVs in the brain areas proximal to the tumor cells due to the strong signals from the tumor cells. We also added discussion on the limitation of human CD63 detection as well as additional references related to other miR-199b's functions.

Additional comments:

1. The authors ought to present a more comprehensive background on the involvement of astrocytes in primary and secondary brain tumors in the introduction or discussion section. Specifically, they should explore the role of astrocytes in the metabolic reprogramming of the tumor microenvironment (TME). This could be achieved by referencing relevant studies such as PMID 29892069, 35899587, 27225120, and 25639230.

Response: Thank you for the comment! We have added these important references to the Discussion (the 3rd paragraph).

2. It would be beneficial for the authors to provide a more extensive context regarding the role of miR-199b in cancer generally, as well as in relation to breast cancer and metastasis. This could be accomplished by citing relevant studies such as PMID 23296799, 35090460, and 32082362, among others.

Response: We have added these additional references to the Discussion (the 1st paragraph).

3. The authors need to be more consistent and clearer about the identity of miR-199b, as miR-199b-5p has a different sequence from miR-199b-3p. They should ensure that they are unambiguously referring to miR-199b-5p throughout the manuscript. Doing so will help avoid any confusion and inaccuracies in interpreting their results.

Response: We have added clarifications that this manuscript refers to miR-199b-5p.

4. It would be advantageous to provide one or two lines of summary or conclusion for the section titled "miR-199b is associated with brain metastasis in BC patients and secreted by brain-tropic MBC cells".

Response: We have added the following sentence at the end of this section: "These results collectively suggest that BC cell-derived, EV-encapsulated miR-199b partakes in the reprogramming of target cells to influence brain metastasis."

5. The authors should incorporate an immunoblot analysis of CD63 in the BBM1 extracellular vesicles (EV) shown in Figure 2B.

Response: CD63 was undetectable in the EVs from BBM1 cells. We have included the negative results in Supplementary Fig. 1b.

Reviewer #4:

In this study, the authors demonstrated that EVs derived from brain-metastatic breast cancer cells shuttled miR-199b-5p to neurons and astrocytes, affecting neural metabolism and promoting brain metastases. Their work is great. I guess that experiments in this study needed very tough work. This manuscript is worth considering for publication in this journal, but it can be considered after addressing the following issues in revision.

Major comment

1. It is interesting that breast cancer cell-derived EVs affect neural metabolism, leading to brain metastases. However, as the authors know, this journal has a high impact on this research field, please emphasize how this idea is novel. They should develop the topic in the discussion section.

Response: We have added a significant amount of discussion to the Discussion section of the revised manuscript. Thank you for the suggestions!

2. In the introduction section, please discuss the relationship between neural metabolites and brain (metastatic) tumor in more detail. Please describe how critical the level of glutamate, glutamine, and lactate for tumor growth.

Response: We have included this in the Discussion (the 2nd paragraph).

3. Please make figure 5(A) understandable. In the experiments using brain slices, it is slightly difficult to understand the time course of the experiments. In some experiments, cancer cells were seeded on the same day (day 2) with EV treatment, but current figure 5(A) doesn't show this time course correctly. Moreover, in figure 5(A), the illustrations of EV treatment and cancer seeding are the same. It is confusing. Please change the illustration.

Response: We have revised Fig. 5a to improve the clarity regarding the time course of cancer cell seeding in some experiments and the illustration.

4. In the experiments using brain slices, they demonstrated that the levels of glutamine and lactate were increased by EV treatment to brain slices, leading to significant cancer cell growth. These data are so amazing. They need to carefully verify the validity of these results. Please check the following points. First, EVs could be also taken up by cancer cells on the brain slice. They need to show data on EV-treated cancer cell growth. They showed data on MDA-231/miR-199b in vitro cell growth in figure 7(C). According to this data, I guess that cancer cell growth is down-regulated by EV treatment, but they need to show clearly. Second, they need to confirm whether the cancer cell growth is changed when the levels of glutamine and lactate in CM are up or downregulated. Thirdly, living cell numbers in the brain slices could be reduced by EV treatment. Current results could be obtained when EV treatment causes cell death and reduce living cell numbers. Finally, SLC transporters can be expressed not only in astrocytes and neurons but also in other types of cells in the brain slices. Please examine and discuss these points.

Response: Thanks for these suggestions. We performed a series of new experiments to strengthen these points. First, we measured the growth of MDA-231 and MDA-231-BR3 cancer cells upon EV treatment. The results showing that the EVs tested here do not significantly affect cancer cell growth are added to new Supplementary Fig. 4a. Secondly, we measured the growth of MDA-231 cancer cells in the CM from differentially treated brain slices and with or without supplementation with glutamine or/and lactate. CM from brain slices treated with high-miR-199b EVs, which contained higher levels of glutamine and lactate, resulted in enhanced cancer cell growth compared to CM from brain slices treated with control EVs. Supplementation of the CM from control EV-treated brain slices with glutamine or lactate to match the levels detected in the CM from high-miR-199-treated groups partially rescue cancer cell growth, whereas adding both glutamine and lactate fully recapitulated the effects of high-miR-199b EVs (new Figure 5g). Thirdly, we measured the viability of brain cells in differentially treated brain slices and showed that EV treatment did not significantly affect brain cell viability (new Supplementary Fig. 4c). Finally, to obtain a comprehensive understanding of the gene expression patterns in different types of brain cells, we re-analyzed a single cell gene expression dataset of human prefrontal cortex samples (GSE168408). Following cell clustering and cell type annotation, five major cell subsets were obtained (new Supplementary Fig. 3a). Expression of EAAT2 (SLC1A2) was the highest in astrocytes, whereas expression of SNAT2 (SLC38A2) and MCT2 (SLC16A7) were the highest in excitatory neurons (new Supplementary Fig. 3b,c). Oligodendrocytes also show some EAAT2 and MCT2 expression; however, in this study we focus on astrocytes and neurons for their metabolic coupling pathway.

5. The authors need to show the bioluminescent image of the body in in vivo experiments (Figure 6). I am interested in the specificity of EV-miR-199b to the brain metastatic tumor. I wonder if EV-miR-199b have some effect on the tumor in other sites such as lung and lymph nodes. If they confirm that EV-miR-199b does not have any effect on lung or lymph nodes, they succeed in demonstrating that EV-miR199b function is specific to brain.

Response: In the spontaneous metastasis model in old Fig. 6F, because the luciferase-labeled cancer cells were injected into the mammary fat pad, the bioluminescent signals from the primary tumors were too intense to allow signals in the brain to be seen and we therefore covered the bodies of mice to focus on the brains instead of

showing whole body imaging. To overcome this shortage and by following the reviewer's suggestion, we generated a new T47D cell line model with miR-199b overexpression and secretion (new Fig. 1d). In a new mouse experiment we first treated mice with i.v. injected EVs, and then performed intracardiac injection using the T47D cell line to assess the effect of EV miR-199b on brain metastasis. The new results showing whole body imaging are added to new Fig. 7g-i and Supplementary Fig. 6. In this model, we observed specifically enhanced metastasis to the brain but not to lung and liver. We also observed metastasis to the ovaries, which could be related to ER signaling in this ER+ tumor model that requires subcutaneous administration of estrogen (estradiol cypionate).

6. The authors should discuss the relationship between BBB and cancer cell-derived EV microRNAs more in detail. Please cite a following paper. Moreover, does miR-199b-5p have some function on BBB? They should confirm at least the difference of BBB penetration efficiencies between MDA-231/miR-199b EV and MDA-231/ctrl EV (Figure 6A).

Tominaga, N., Kosaka, N., Ono, M. et al. Brain metastatic cancer cells release microRNA-181c-containing extracellular vesicles capable of destructing blood–brain barrier. *Nat Commun* 6, 6716 (2015).

Response: We have added this important reference to the Discussion (at the beginning of the 2nd paragraph). Quantification of the percentages of brain cells taking up intravenously injected EVs with high or low miR-199b-5p showed no significant difference (new Fig. 6a and Supplementary Fig. 5).

Minor comment

0. Please show how to label cancer cells by mCherry in the methods section.

Response: We have added the method to fluorescently label the cells (by using the pEZX-AM03 lentiviral vector for mCherry and pEZX-MR03 lentiviral vector for GFP) to the Methods section.

1. Please show the scale bar on the left picture in figure 5(B).

Response: The scale bar has been added.

REVIEWERS' COMMENTS

Reviewer #1 (Remarks to the Author):

This is a comprehensive and exciting revision, congratulations to the authors. The data now support the hypothesis.

Reviewer #2 (Remarks to the Author):

The authors have addressed my previous points. I don't have additional concerns.

Reviewer #4 (Remarks to the Author):

The current manuscript has been revised well. I recommend that it be accepted for publication.

Response to Reviewers

The reviewers did not raise any additional comments. We sincerely thank all reviewers for the constructive comments and suggestions that have helped us improve our manuscript!